



# Assessment of Severe Aerosol Events from NASA MODIS and VIIRS Aerosol Products for Data Assimilation and Climate Continuity

Amanda Gumber[1], Jeffrey S. Reid[2], Robert E. Holz[1], Thomas F. Eck[3,4], N. Christina Hsu[4], Robert C. Levy[4], Jianglong Zhang[5], Paolo Veglio[1]

[1]Space Science and Engineering Center, University of Wisconsin-Madison, Madison, WI 53706, USA
[2]U.S. Naval Research Laboratory, Monterey, CA 93943, USA
[3]Goddard Earth Sciences Technology and Research (GESTAR) II, University of Maryland Baltimore County, Baltimore, MD 21250, USA
[4]NASA Goddard Space Flight Center, Greenbelt, MD 20771, USA
[5]Department of Atmospheric Sciences, University of North Dakota, Grand Forks, ND 58202, USA

*Correspondence to*: Amanda Gumber (amanda.gumber@ssec.wisc.edu)

**Abstract.** While the use and data assimilation (DA) of operational Moderate Resolution Imaging Spectroradiometer (MODIS) aerosol data is commonplace, MODIS is scheduled to sunset in the next year. For data continuity, focus has turned to the development of next generation aerosol products and sensors such as those associated with the

Visible Infrared Imaging Radiometer Suite (VIIRS) on Suomi NPOESS Preparation Project (S-NPP) and NOAA-20. Like MODIS algorithms, products from these sensors require their own set of extensive error characterization and correction exercises. This is particularly true in the context of monitoring significant aerosol events that tax an algorithm's ability to separate cloud from aerosol and account for multiple scattering related errors exacerbated by uncertainties in aerosol optical properties. To investigate the performance of polar orbiting satellite algorithms to

monitor and characterize significant events a Level 3 (L3) product has been developed, using a consistent aggregation methodology, for four years of observations (2016-2019). Included in this product is AErosol RObotic NETwork (AERONET), MODIS Dark Target, Deep Blue, and Multi-Angle Implementation of Atmospheric Correction (MAIAC) algorithms. These MODIS "baseline algorithms" are compared to NASA's recently released NASA Deep Blue algorithm for use with VIIRS. Using this new dataset, the relative performance of the algorithms

for both land and ocean were investigated with a focus on the relative skill of detecting severe events and accuracy of the retrievals using AERONET. Maps of higher percentile AOD regions of the world by product, identified those with the highest measured AODs, and determined what is high by local standards. While patterns in AOD match across products and median to moderate AOD values match well, there are regionally correlated biases between products based on sampling, algorithm differences, and AOD range-in particular for higher AOD events. Most

notable are differences in Boreal biomass burning and Saharan dust. Significant percentile biases that must be accounted for when data is used in trend studies, data assimilation, or inverse modeling. These biases vary by aerosol regime and are likely due to retrieval assumptions on lower boundary condition and aerosol optical models.



# 1 Introduction

Monitoring the aerosol system is an integral part of many applications such as air quality, human health, climate monitoring, and visibility impairment. Passive imager observations, from polar orbiting, sun synchronous satellites, have allowed researchers to monitor global aerosol for decades. With once-a-day or better coverage over most areas of the globe, these satellite sensors have helped to characterize overall aerosol climatology, but also to detect significant events of dust, smoke, and pollution. From such data, research frequently maps regional impacts and

estimate emissions. More recently, the incorporation of satellite aerosol data into data assimilation (DA) systems has worked to systematize inverse estimations of emissions, improve aerosol forecasting, and open numerous new climate applications (2D-Var- Zhang et al., 2008; 3D-Var-Randles et al., 2017; 4D-Var-Benedetti et al., 2009; Ensemble Kalman Filter-Schutgens et al., 2010; Khade et al., 2012; Pagowski and Grell, 2012; Rubin et al., 2016; 2017; Hybrid-Schwartz et al., 2014). Model reanalyses that depend heavily on consistent and well-characterized

satellite datasets in their assimilation and evaluation cycles are used by the community to establish trends and estimate emissions (Lynch et al., 2016; Randles et al., 2017; Yumimoto et al., 2017; Inness et al., 2019). Yet, although satellite data is continuing to improve, bias and uncertainty remain due to instrument calibration and retrieval method shortcomings (Zhang and Reid, 2006; Zhang and Reid, 2010; Shi et al., 2011; Sayer et al., 2013; Levy et al., 2018). This is especially true for observing and quantifying severe events, which is the focus of this

manuscript. Indeed, the transition from the Terra and A-Train sensors to the Joint Polar Satellite System is happening just as climate changed significant events such as wildfires are on the rise (Bondur et al., 2020; Coogan et al., 2020; Zhang et al., 2020).

In addition to simple stochastic errors, satellite products show strong and spatially correlated errors that systematically vary with aerosol optical depth (AOD), composition, and lower boundary (e.g., non-aerosol = surface

reflectance, molecular scatterings and absorptions, clouds) conditions (Shi et al., 2011). Both inverse modeling and Data Assimilation (DA) systems are sensitive to observational errors, thus requiring significant quality assurance corrections and careful filtering of the satellite data (Zhang et al., 2008; Hyer et al., 2011).

Examples of satellite-derived datasets used for DA or inverse-modeling include aerosol retrievals from the Moderate Resolution Imaging Spectroradiometer (MODIS), on Terra and Aqua (Sessions et al., 2015; Xian et al., 2019) and

more recently from the Visible Infrared Imaging Radiometer Suite (VIIRS) on the Suomi-National Polar-orbiting Operational Environmental Satellite System Preparation Project (S-NPP) and JPSS-1 (now known as NOAA-20) satellites. While other satellite products have been used in data assimilation, including the Cloud-Aerosol Lidar with Orthogonal Polarization (CALIOP) and Multi Angle Imaging Spectroradiometer (MISR) (Sekiyama et al., 2010; Zhang et al., 2011; Lynch et al., 2016), product lines associated with MODIS and VIIRS see the most usage due to

their coverage, accessibility, delivery speed, and high level of characterization (see requirements outlined in Zhang et al., 2014; Benedetti et al., 2018). Given the imminent transition from MODIS to VIIRS based global observations, the focus is on their derived products of total aerosol optical depth (AOD).

Since quality-assured data are needed for aerosol DA, it is important to identify and characterize biases and uncertainties for products from specific instruments and retrieval algorithms. In addition, for "severe aerosol events"

which occur relatively infrequently and may be defined differently based on application and location one must be



extremely careful in assessing uncertainties. In other words, characterizing *outlier* uncertainty may be an entirely different exercise than characterizing *bulk* uncertainty.

The operational MODIS and VIIRS-derived AOD datasets are intended to represent aerosol conditions in clear-sky (non-cloudy) conditions, and over land and ocean surfaces which are free of ice/snow, glint, and underwater

sediments. Therefore, their bulk uncertainty is related to 1) estimation of the lower boundary condition (e.g., surface reflectance plus Rayleigh/molecular scattering and absorption), 2) assumption of an aerosol model (e.g., physical and optical properties) which is sufficiently representative of the aerosol in the scene, and 3) masking of clouds, ice/snow and other retrieval "obstacles".  For the lowest AOD conditions, it is relatively easy to separate aerosols from clouds and choice of aerosol model is relatively unimportant. Therefore, for low AOD, aerosol retrieval

uncertainties are dominated by uncertainties in the surface boundary condition. As AOD increases, choice of aerosol model becomes more important, as errors in assumed single scattering albedo (SSA), size or shape distributions, and interactions of multiple scattering lead to increased error '(Shi et al., 2019). At even larger AODs, opaque aerosol begins to look like clouds or other retrieval obstacles. In fact, for the largest (extreme) AODs, the algorithm may mask these scenes entirely leading to no retrieval at all.

A challenge facing satellite aerosol data product development is that there is no fundamental spatially contiguous dataset to provide validation; everything is typically inferred from point measurements. While field experiments often provide a high frequency of observations over a small area, most of the validation relies on using ground-based measurements from the AErosol RObotic NETwork (AERONET; Holben et al., 1998). AERONET consists of globally distributed sun photometers capable of providing near-real-time AOD, size and absorption data has become

the validation standard throughout the satellite aerosol community to benchmark products and identifying biases (Zhang and Reid, 2006; Hyer et al., 2011; Shi et al., 2011; Sayer et al., 2013; Sayer et al., 2018). AERONET measurements are also incorporated into aerosol forecasting models despite the limited amount of spatial coverage they have (Schutegens et al., 2010; Randles et al., 2017; Rubin et al., 2017).

While providing a global benchmark dataset with numerous sites, using AERONET to investigate severe aerosol

events is limited due to the inherent sparse nature of AERONET coverage where AERONET rightly observes low AOD, but nearby thick aerosol plume features exist (or vice versa – AERONET manages to sample an extremely localized plume). AERONET provides an overall strict cloud screening, meaning that satellite colocations with such ground-based data are essentially doubly cloud-screened, leading to an under-estimate of satellite-retrieval uncertainty. Nevertheless, while cloud screening has been vastly improved, AERONET level 2 products can still

screen out haze conditions (Eck et al., 2018). For the very highest AOD conditions, the solar disk can be so attenuated that mid-visible AOD measurements are no longer possible. Regardless, with these caveats and performing the work in proper context, AERONET is still the most reliable sensor for providing verification if placed in the proper context, especially over larger scales.

This is the first report of several studies that investigate the nature and trend in severe aerosol events in remote

sensing data and modeling simulations. The first task is to answer the question "What constitutes a severe aerosol event?" That is, what is the measured median AOD versus 84[th], 95[th] or 98% events over the globe? In this article, a baseline of the global distribution of higher-percentile AOD retrievals from NASA's polar orbiting MODIS and



VIIRS aerosol products is developed, and the resulting probability distributions and biases are examined. The
MODIS and VIIRS instruments are the basis for the multi decadal NASA aerosol climate data records and are the

most applied products used for model evaluation and DA. Also of interest is product performance as the community
transitions of the EOS MODIS to the JPSS VIIRS era and what this implies for monitoring trends in significant
events in association with climate change. The focus is on the bulk relative probability distributions of high AOD
events and pairwise relationships between products. However, since the most-severe aerosol events occur near their
terrestrial sources over land, statistics are developed that define severe events locally.

While many aerosol product evaluation studies have been conducted (Remer et al., 2008; Levy et al., 2010; Hyer et
al., 2011; Sayer et al., 2013; Li et al., 2014; Bilal et al., 2018; Lyapustin et al., 2018; Wei et al., 2019; Reid et al.,
2022), this study deviates in that it does not strictly compare products against AERONET, or even between each
other, but rather an assessment of differences in the probability distribution functions of AOD with a focus on the
relative differences between the aerosol retrieval algorithms. Section 2 describes each of the satellite data products,

the ground based AERONET observation, and the software used to grid and aggregate the data. Section 3 provides
both a global overview of the satellite datasets describing the aerosol PDFs and characterizing the nature of severe
events. Section 4 identifies and investigates regions associated with high aerosol loading using satellite datasets and
using AERONET data provide a quantitative assessment of the retrieval biases of severe events. The discussion and
conclusions of this study are finalized in Section 5.


## 2 Data

The datasets in this study use the benchmark AOD values at 0.55mm ($AOD_{550}$). Here a 4-year time span is used
from 2016-2019 to create a L3 product designed after commonly applied DA products gridded at 1-degree x 1-
degree using a consistent aggregation method. From each dataset the highest quality assurance flag available is used

to approximate the quality of assimilation-grade data without additional filtering.

### 2.1 AERONET

The federated AErosol RObotic NETwork (AERONET; Holben et al., 1998) network of Cimel sun sky radiometers
is the primary basis set for evaluating satellite products. The nature of this data is discussed in detail in Giles et al.,

(2019) and does not need to be repeated here. Over the 4-year time span, 261,255 AERONET AOD observations are
collected from 102 to 231 sites each day. To match typical DA cycles, all AERONET data is sampled globally and
averaged at 6-hour time interval for each file. While this study uses Version 3 Level 1.5 AOD data, it is noteworthy
that these products now share the same cloud screening criterion and products are regularly updated to final
calibrations as instruments are recalibrated. Thus, they are much more similar to level 2 than in the past. Since

AERONET sun photometers do not observe AOD at the 550 nm wavelength, the Spectral Deconvolution Algorithm

(SDA) is applied to the five 380, 440, 500, 675, and 870 nm channels to derive $AOD_{550}$ (O'Neill et al., 2003; 2008; Kaku et al., 2014). SDA is also used in the analysis to isolate fine and coarse mode AODs from AERONET.

## 2.2 Satellite products

This study focuses on the transition and consistency of Terra and Aqua based MODIS to S-NPP and JPSS VIIRS products with the data quality assessed in the context for data assimilation. The polar orbiting sun-synchronous morning Terra and afternoon Aqua satellites were launched in 1999 and 2002, respectively. Since both Aqua and S-NPP are both in afternoon orbit, this study only focuses on Aqua-MODIS.

For MODIS, this study examines products from three of NASA's aerosol retrieval algorithms. Two of them, known

as Dark Target (DT) and Deep Blue (DB), are contained together in a product known as MYD04. DT retrieves aerosols over ocean and vegetated (dark) land, whereas DB retrieves over vegetated and barren (brighter) surfaces. Although derived separately, DT and DB are also combined into a joint product within the MYD04, and that DT/DB product is used in this study. It is important to note that both DT and DB are performing instantaneous single-view granule-based retrievals, meaning that there is no information used from previous or subsequent granules. On the

other hand, the Multi-Angle Implementation of Atmospheric Correction (MAIAC) utilizes multiple overpasses to derive a higher resolution 1 km product with simultaneous AOD and land surface products. MAIAC products are contained in a product known as MCD19, and they are derived using combinations of MODIS observations from both Terra and Aqua. The latest version of MCD19 is known as Collection 6.

For VIIRS, a product known as AERDB is examined, which follows the heritage of DB on MODIS. Unlike on

MODIS, where DB is performed over land only, DB also uses an algorithm known as Satellite Ocean Aerosol Retrieval (SOAR) to retrieve over ocean. This study uses Version 1.0 of the AERDB product. Note that now there is now an available version of DT on VIIRS (known as AERDT; Sawyer et al., 2020), but was not yet operational at the commencement of this study.

All products include an estimation of total AOD at 0.55 μm ($AOD_{550}$) as well as spectral AOD at selected

wavelength bands. While DT on MODIS reports fine and coarse AOD over ocean, no other product reports that parameter over land.

### 2.2.1 MODIS Combined Dark Target Deep Blue



MODIS contains 36 spectral bands that range over 0.4–14.4 μm with spatial resolutions varying from 250 m to 1

km, depending on the selected band, with a swath of 2330 km (cross track) x 10 km (along track at nadir). The

MODIS Dark Target (DT) algorithm is the heritage aerosol algorithm used for global aerosol monitoring. DT

generates $AOD_{550}$ products over visually dark surfaces such as vegetated land and ocean regions, using two separate

algorithms (Kaufman et al., 1997; Levy et al., 2013). The ocean algorithm uses a lookup table approach based on

fine and coarse mode aerosol models to compute modeled reflectance from six MODIS observed wavelengths (0.55,

0.65, 0.86, 1.24, 1.63, and 2.11 μm). The land algorithm uses a similar lookup table-based approach from

precomputed radiative transfer calculation for surface and aerosol parameters, but only uses 3 spectral wavelengths

(0.47, 0.65 and 2.1 μm). Retrievals are based on the aggregates of NxN worth of native-resolution pixels, where the

N equals 40, 20 or 10 depending on the native resolution, resulting in nadir retrieval sizes ranging from 10x10 km

(at nadir) to ~50x30km (edge of swath). Numerous studies have evaluated DT's performance from inception of

collection 3.1 until its current version of collection 6.1 with evaluations provided in Levy et al. (2013), Sayer et al.

(2013), Sayer et al. (2017), and Wei et al. (2018).  Collection 6.1 has significantly removed many of the previous

deficiencies, such as insufficient cloud screening, better aerosol-cloud discrimination, and improvements in

constraining the lower boundary condition.

A limitation of the DT algorithm is its inability to retrieve AOD over bright desert surfaces due to the loss of

contrast to isolate the aerosol signal. The MODIS Deep Blue (DB) algorithm was initially developed to better

retrieve $AOD_{550}$ over bright desert surfaces, taking advantage of the fact that iron in sand and soil absorbs blue light

and thus reduces the surface albedo. In other words, deserts appear "dark" at blue and deep blue (e.g., 0.41 μm)

wavelengths to provide sufficient contrast for aerosol retrieval. Since vegetation also appears dark in deep blue

wavelengths, the DB algorithm has been subsequently expanded to also include vegetated surfaces. Like DT, the

MODIS DB algorithm is only performed over snow-free and cloud-cleared land pixels, however, instead uses top-

of-atmosphere reflectance at 650, 470, and 412 nm to determine spectral $AOD_{550}$ (Hsu et al., 2004; Hsu et al., 2013)

by matching to lookup tables. Also, like DT, the DB product for MODIS is provided at a nadir spatial resolution of

10x10 km and edge of swath 50x30 km. Collection 6.1 contains improvements for heavy smoke detection,

heterogeneous terrain, elevated surface types, and changes within aerosol optical models (Sayer et al., 2019).

Given the different use cases of the DT and DB algorithms, the MODIS combined product provides a retrieval-by-

retrieval selection from both algorithms to form a merged dataset that is recommended for general use by both the



DT and DB development teams. Selection is based on the underlying surface's monthly-averaged Normalized

Difference Vegetation Index (NDVI) value gleaned from a separate MODIS product (Levy et al., 2013). So, just as

there have traditionally been over ocean and over land retrievals, the current paradigm is to likewise have land

retrievals separated by lower boundary condition with future releases even able to retrieve over snow and ice. Using

the combined Dark Target and Deep Blue product increases spatial coverage, especially over deserts and low

vegetation regions. This combined DT/DB methodology is most comparable to the Deep Blue retrieval applied to

VIIRS (see below) and thus is the focus of this analysis.

**2.2.2 MAIAC**

The MODIS Multi-Angle Implementation of Atmospheric Correction (MAIAC) product uses time-series analysis

and a combination of image-based and pixel-base processing (Lyapustin et al., 2011, Lyapustin et al., 2018).

MAIAC grids MODIS L1B data to a 1km resolution and creates a 16-day time series using a sliding window

technique in order to obtain multiple viewing angles to capture the surface Bidirectional Reflectance Distribution

Function (BRDF). This time series analysis separates slowly varying lower boundary conditions from more rapid

atmospheric conditions to provide a 1x1 km product projected onto a sinusoidal grid. While MAIAC is generated for

overland pixels, it also captures coastal waters and major island areas.  The highest quality assurance of the product

only retrieves AOD over land and land-containing regions.

There have been several regional evaluation studies of the MAIAC algorithm (Martins et al., 2017; Superczynski et

al., 2017; Chen et al., 2021). However, to our knowledge this is first multiyear global analysis.

**2.2.3 VIIRS Deep Blue**

VIIRS on S-NPP and NOAA-20 has 22 spectral bands ranging from 0.412 μm to 12.01 μm. While both MODIS and

VIIRS have a fixed Field of View (FOV), VIIRS data is aggregated on board to provide a more consistent spatial

resolution across the swath with a nominal surface footprint of 750 meters for the M band channels that are used by

the algorithms (Sayer et al., 2017). The sensor has a swath width of 3060 km, allowing for complete global coverage

over a day including the equator where MODIS has gaps. A version of the Deep Blue algorithm that produces

aerosol AOD and fine mode fraction at a nadir spatial resolution of 6x6 km for both dark scenes and bright land

surfaces was selected by NASA as the primary Earth System Data Record (ESDR) for VIIRS. Over land, Deep Blue



draws its heritage from the MODIS-based Deep Blue product of Hsu et al., (2013; 2019) while over water the

product uses the Satellite Ocean Aerosol Retrieval (SOAR; Sayer et al., 2017;2018). SOAR has heritage based on

SeaWiFS aerosol products and uses a traditional least-squares fit of multiple channels to retrieve $AOD_{550}$ (Sayer et

al., 2010).

### 2.3 L3 Data Integration through YORI


For the purposes of global model applications, data is aggregated and regridded to 1x1 degree resolution suitable for

most global aerosol data assimilation applications using a python toolkit developed by the UW A-SIPS called Yori

that provides a consistent framework to integrate multiple products into a gridded dataset known as Level 3 (L3).

Yori provides easy integration of new datasets from both satellite and ground-based observations allowing for

custom filtering and masks as well as multi-dimensional histograms for each grid cell.  Currently, Yori is used for

the L3 cloud products for VIIRS and is being integrated into the MODIS processing for Collection 7 to generate the

L3 cloud and aerosol MYD08 products. That is, the regrading tool used in this study is the same one used

operationally at NASA for its operational aerosol products. The products for this analysis are gridded at 1-degree x

1-degree every 6 hours to match the standard output of the global aerosol models. The 6-hourly L3 files are then

aggregated to the longer time domains used in this investigation. Gridded Yori files are plotted globally, but with

more in-depth regional investigations for the region that show a high spread in AOD values.

Figure 1 (a) provides a map of AERONET sites used in this analysis and the subdomains for more in-depth

discussion. Figure 1(b), (c) and (d) provides maps of the number of aggregated 1x1 degree data points for MODIS

Aqua DT/DB, MAIAC and VIIRS, respectively, generated over the 2016-2019 study period. Clearly, there are

significantly different numbers of samples by a product for any given region, largely dictated by cloud cover and for

over ocean, sun glint and available daylight. This is highlighted further in Fig. 1(e)-(g) where the ratios of the

number of data points between products is provided. For example, for the same Aqua MODIS swath, MAIAC

provides a 10-20% higher data population when upscaled to the 1x1 degree grid. This is because with its 1 km

uniform resolution and Boolean cloud flag nature, there is likelihood of generating data somewhere within the 1°x1°

grid. The VIIRS DB product, with its 30% wider swath, geometrically out-samples the Aqua MODIS counterpart by

a likewise amount. Interestingly, overland Aqua MODIS MAIAC out-samples VIIRS in some regions by as much as

a factor of two, again due to its much-higher 1 km resolution.



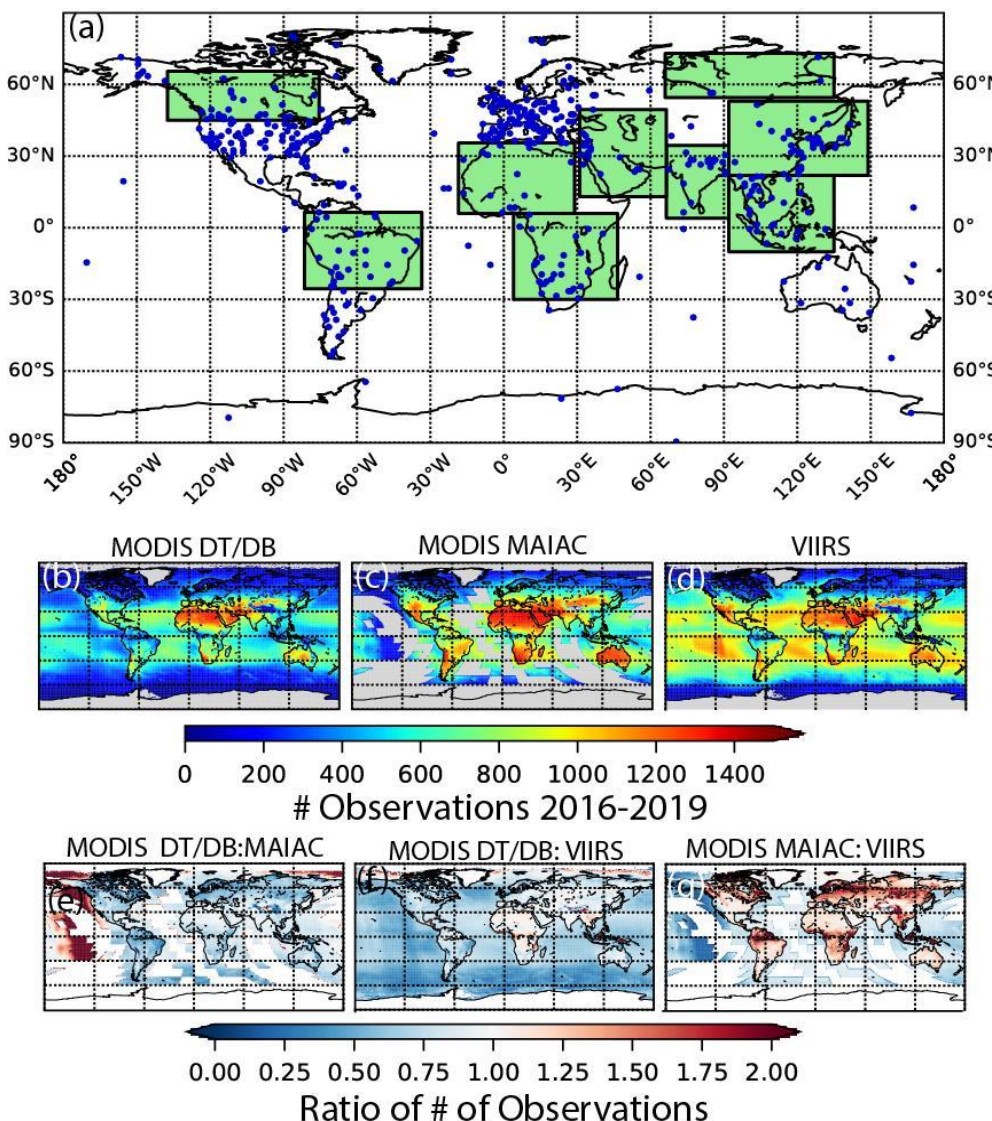

**Figure 1.** (a) Selected Regions for more detailed analysis of severe events including Boreal Canada, South America, Saharan Africa, central Africa, southwest Asia, India, eastern Asia, boreal Asia, and Southeast Asia. Included are the AERONET locations use in this analysis. (b)-(d), The number of 1x1 degree data points used over the 4-year study period for MODIS DT/DB, MODIS MAIAC, and VIIRS. (e)-(g), Ratios of the number of data points between products.



## 3 Global Overview


There are many metrics available to intercompare satellite products. Here two approaches are taken, a global analysis of the differences in product probability distribution functions, and pairwise comparisons by regression. These are done as a global survey (Section 3) followed by more in-depth regional analysis and discussion (Section 4). In this subsection, the relative differences in the global distribution of aerosol events across MODIS and VIIRS

algorithms are globally mapped by probability distribution (Section 3.1) followed by mapped regressions in a manner similar to Shi et al., (2011) in Section 3.2. Gridded netcdf files of this data in these figures are provided as supplemental materials.

### 3.1 Event probabilities


The global distribution of 2016 through 2019 aggregated $AOD_{550}$ datasets is provided in Fig. 2 projecting $AOD_{550}$ onto a lognormal distribution. Provided for each grid cell are the median (i.e., 50th percentile), 84th percentile, and +1 geometric standard deviation ($\sigma_g$, here taken as the ratio of the 84th percentile to the median). Higher-level percentiles (95% and 98%) as well as the number of days with $AOD_{550}>0.8$ are provided in Fig. 3. Ratios of the MODIS products to VIIRS are also included in the lower two panel sets in Fig. 2 & 3. VIIRS DB is chosen as the

baseline for the ratios since it is the newest of the products and provides an integrated bright and dark surface retrieval. Over land and ocean ratios are also provided in Table 1, with additional regional values provided for discussion in Section 4. The 84th percentile values accounts for that value that separates the highest ~8 weeks of loading for the year, and its ratio to the median (an approximation of $\sigma_g$) is a measure of the relative dispersion of the $AOD_{550}$ probability distribution. The 95th and 98th percentile values of $AOD_{550}$ (e.g., highest 18 and 7 days) focus

on the tail end of the distribution which comprise the most significant severe events. Finally, the number of days with $AOD_{550}>0.8$ metric provides an absolute threshold of the world's most significant aerosol hotspots. These metrics where selected to provide insight to where the most significant aerosol events occurred, what is considered locally an exceptional event, and how these events vary spatially between datasets. It also shows areas where datasets do not make retrievals such as over the Arctic and over most of the ocean for MODIS MAIAC, which

contains a quality flag requiring a portion of land within the sinusoidal gridded area. It is important to highlight that given the lack of a truth data set for the global results, it is not possible to directly screen for incorrect cloud/aerosol classification.



**Figure 2.** Median (left), 84th percentile (center), and geometric standard deviation (right) of AOD for VIIRS Deep Blue, MODIS Dark Target/Deep Blue combined, and MODIS MAIAC. The bottom two rows present the ratio of MODIS DT/DB and MODIS MAIAC relative to VIIRS DB over the years 2016-2019 for each individual 1 degree by 1 degree area. The ratios filter areas where 84th percentile of AOD < 0.1.





**Figure 3.** 95th percentile of AOD (left), 98th percentile of AOD (center), and number of days where AOD > 0.8 (right) for VIIRS DB, MODIS DT/DB, and MODIS MAIAC for 2016-2019. The ratio of MODIS DT/DB and MODIS MAIAC to VIIRS DB is presented in the bottom two rows.


| Region | Satellite | Median | 84th Percentile | $\sigma_g$ | 95th Percentile | 98th Percentile |
|--------|-----------|--------|-----------------|------------|-----------------|-----------------|
| Global | MODIS DT/DB | 0.158 | 0.403 | 2.55 | 0.691 | 0.960 |
| Land | MODIS MAIAC | 0.147 | 0.324 | 2.20 | 0.574 | 0.833 |
| | VIIRS DB | 0.151 | 0.414 | 2.74 | 0.733 | 1.041 |


| | | | | | | |
|---|---|---|---|---|---|---|
| Global Ocean | MODIS DT/DB | 0.114 | 0.217 | 1.91 | 0.360 | 0.517 |
| | MODIS MAIAC | 0.109 | 0.220 | 2.01 | 0.351 | 0.468 |
| | VIIRS DB | 0.114 | 0.203 | 1.78 | 0.313 | 0.450 |
| Boreal Asia | MODIS DT/DB | 0.131 | 0.355 | 2.72 | 0.994 | 1.807 |
| | MODIS MAIAC | 0.113 | 0.235 | 2.07 | 0.565 | 1.412 |
| | VIIRS DB | 0.097 | 0.293 | 3.04 | 0.841 | 1.429 |
| Boreal North America | MODIS DT/DB | 0.109 | 0.263 | 2.42 | 0.548 | 0.942 |
| | MODIS MAIAC | 0.105 | 0.214 | 2.05 | 0.333 | 0.553 |
| | VIIRS DB | 0.083 | 0.219 | 2.62 | 0.539 | 1.353 |
| Central Africa | MODIS DT/DB | 0.228 | 0.502 | 2.20 | 0.788 | 1.022 |
| | MODIS MAIAC | 0.201 | 0.380 | 1.89 | 0.612 | 0.812 |
| | VIIRS DB | 0.221 | 0.542 | 2.45 | 0.885 | 1.131 |
| South America | MODIS DT/DB | 0.130 | 0.274 | 2.11 | 0.459 | 0.672 |
| | MODIS MAIAC | 0.129 | 0.229 | 1.77 | 0.325 | 0.409 |
| | VIIRS DB | 0.125 | 0.279 | 2.23 | 0.476 | 0.683 |
| Southeast Asia | MODIS DT/DB | 0.197 | 0.406 | 2.06 | 0.688 | 0.960 |
| | MODIS MAIAC | 0.188 | 0.355 | 1.89 | 0.603 | 0.870 |
| | VIIRS DB | 0.201 | 0.389 | 1.94 | 0.650 | 0.939 |
| Saharan Africa | MODIS DT/DB | 0.261 | 0.538 | 2.06 | 0.828 | 1.120 |
| | MODIS MAIAC | 0.238 | 0.473 | 1.99 | 0.761 | 1.043 |
| | VIIRS DB | 0.336 | 0.680 | 2.02 | 1.080 | 1.466 |
| Southwest Asia | MODIS DT/DB | 0.247 | 0.489 | 1.98 | 0.730 | 0.942 |
| | MODIS MAIAC | 0.181 | 0.375 | 2.07 | 0.637 | 0.883 |
| | VIIRS DB | 0.243 | 0.513 | 2.11 | 0.812 | 1.105 |
| South Asia | MODIS DT/DB | 0.368 | 0.729 | 1.98 | 1.126 | 1.479 |
| | MODIS MAIAC | 0.407 | 0.747 | 1.84 | 1.078 | 1.372 |
| | VIIRS DB | 0.383 | 0.674 | 1.76 | 0.937 | 1.157 |
| Eastern Asia | MODIS DT/DB | 0.150 | 0.444 | 2.96 | 0.802 | 1.125 |
| | MODIS MAIAC | 0.172 | 0.392 | 2.28 | 0.690 | 0.989 |
| | VIIRS DB | 0.135 | 0.400 | 2.98 | 0.750 | 1.066 |

**Table 1.** Median, 84th percentile, and +1 geometric standard deviation ($\sigma_g$) for selected regions for each dataset. Apart from Global Ocean, all values are for over land only.

As expected, the overall distribution of median $AOD_{550}$ is consistent with existing aerosol climatologies of aerosol means by satellite (Mishchenko et al., 2007; Remer et al., 2008; Li et al., 2009; Wei et al., 2019; Sogacheva et al., 2020) and operational model (Sessions et al., 2015; Xian et al., 2019). Globally the median over land $AOD_{550}$ is consistent between products, ranging from 0.14-0.15 (Table 1). Median $AOD_{550}$ were in agreement by product and were highest in the subtropical belt of a) South Asia Indo-Gangetic plain, $0.37<AOD_{550}<0.41$, for pollution and biomass burning coupled with haze formation (Dey and Di Girolamo, 2011); b) Saharan Africa, $0.25<AOD_{550}<0.35$, for dust (Caton-Harrison et al., 2019); c) tropical and central Africa $0.20<AOD_{550}<0.23$ for smoke (Swap et al., 2003; Eck et al., 2013); d) SW Asia, $0.18<AOD_{550}<0.25$, for a combination of dust and pollution (Reid et al., 2013; Al-Taani et al., 2019). Additional sub domain hotspots in median $AOD_{550}$ include portions of the North China Plain (An et al., 2019) and the Taklimakan desert (Ge et al., 2014). Over ocean, MODIS DT/DB and VIIRS are within 0.01 of each other (0.10-0.12). By region, however, spatially correlated biases between products are readily apparent. For the ratios of MODIS products to VIIRS in Fig. 2 values greater than 1 indicates the MODIS products





observed higher $AOD_{550}$ values, whereas values less than 1 indicates VIIRS DB observed higher AOD values. As reflected in the domain average, VIIRS DB overall shows the highest climatological magnitude of gridded $AOD_{550}$ medians while MODIS MAIAC has the lowest magnitudes. The most notable locations of differences are clean background (low AOD) regions of the arid western United States, the Gobi Desert, and the arctic where both

MODIS products are higher than VIIRS. Although at the median level, this is only a 0.03 difference in AOD. For more heavily loaded environments, the largest discrepancies are in central to southern Africa and East Asia. In Section 4, these regional differences are investigated using AERONET to help quantify the biases.

At the 84th% level, which inherently accounts for some seasonality in aerosol loadings, additional aerosol hot spots are visible that defined the remainder of the regions in Fig. 1. Most notably are the biomass burning regions of South

America, Southeast Asia, and the boreal Asia and Canada as well as a more consistent identification of the Taklimakan desert. At the 84th% level, $AOD_{550}$ signal is good, and products also largely agree, but some divergence becomes evident. For example, over land at the 84th percentile level, MAIAC provides distinctively lower AODs for nearly all regions (0.31 for MAIAC versus ~ 0.4 for MODIS DT/DB and VIIRS). The lowest values for MAIAC are associated with central African and boreal burning. While MODIS DT/DB and VIIRS are largely within 10% of

each other, for Saharan Africa VIIRS is 25% higher at the 84th percentile. By region, the strongest divergence between satellites is in central to southern Africa-unsurprising given the variability in aerosol speciation, single scattering albedo and land surface characteristics. Divergence also still exists in the western United States, although even at the 84th level, AODs are still quite low. Finally, the sign of the ratios to VIIRS often switches between land and water-an indicator of algorithmic differences used for those two surfaces.

In the context of AOD variability, $\sigma_g$ normalizes the 84th% with respect to the medians of each dataset to make them more comparable. $\sigma_g$ can be used as an indicator of $AOD_{550}$ dispersion, with higher values indicative of a higher prevalence of episodic aerosol events relative to the mean. Over ocean, the equal area average value is ~1.8-2.0, with notable enhancements associated with the northern portion Saharan dust plume (due to seasonal variability) as well as northern latitudes, presumably due to biomass burning events. Over land, regions with the highest spread (in

excess of 3) include boreal biomass burning of Siberia and North America followed by seasonal burning regions of Africa, South America and Southeast Asia. Datasets are largely consistent in region identification to these hotspots. For dust, the Taklimakan desert and coastal Argentina are also highlighted. Interestingly, the distribution spread for African dust, is much more muted-likely owing to the dominance of a single and frequently active dust source.

Notable differences between datasets are apparent in Fig. 2, especially between land and ocean. Overall, MODIS

MAIAC, when aggregated, shows the lowest values of dispersion in comparison to the other two datasets. Regions that show high standard deviations for VIIRS DB and MODIS DT/DB include Northwest North America, Southern Africa, Central South America, Southeast Asia, Western China, and Central Russia. These all are associated with biomass burning events.

To further evaluate the differences between each of these datasets from a severe events perspective, Fig. 3 includes

the 95th and 98th percentile values of $AOD_{550}$ (e.g., highest 18 and 7 days). Over the ocean, MODIS DT/DB generally observes larger 95th percentile values of $AOD_{550}$ compared to VIIRS DB. The difference in products is especially noticeable over the central Atlantic where dust events occur, as well as high-mid latitudes and Arctic.



Over land, both MODIS products show greater values of 95th percentile $AOD_{550}$ than VIIRS over India, Eastern Asia, southwest North America, western coast of South America, and South Africa. VIIRS DB dominates over land

with higher values over Boreal Canada, central South America, Saharan Africa, and Boreal Asia. In the case of the Arctic Ocean, Africa and Southwest Asia a clear transition from a low MODIS to VIIRS ratio to high from the boreal to the Arctic Ocean, is likely related to the switch between land to ocean retrievals. 98th percentile values of $AOD_{550}$ have similar spatial patterns to the 95th percentile of $AOD_{550}$. MODIS DT/DB and VIIRS DB show a large plume off the western coast of the Hawaiian Islands.

As opposed to probability distributions, the number of days with $AOD_{550} > 0.8$ presented in Fig. 3 was used in this analysis as a threshold benchmark. While probability distributions to an extent normalize out sampling (account for both swath width and for over ocean, the higher swath fraction to glint), threshold scores are useful in their ability to detect an event. $AOD_{550} > 0.8$ is used because the lowest global over land 98th percentile value of all datasets is approximately 0.8 (0.82 for MAIAC) and corresponds to an AOD alert as part of the ICAP_MME consensus

(Sessions et al., 2015). While thresholds are useful, they can also be problematic given the overall lognormal distribution of AOD; slight systematic biases may result in larger systematic differences in a threshold metric. VIIRS, with its wider swath, could be expected to observe higher AOD events than MODIS. However, over land the differences between MODIS and VIIRS largely resemble the differences in the 95th and 98% AODs. Strong gradients in the ratios from land to water further highlight the effect of having different ocean and land retrievals.

Thus, retrieval differences for VIIRS overtake the gains made by coverage for this type of metric. This is explored further in the pairwise comparisons conducted in Section 3.2.

 Taken as a whole, the over land VIIRS DB dataset, with its larger swath, shows the most $AOD_{550} > 0.8$ days in comparison to both MODIS datasets over Africa, Boreal Canada, central South America, and Boreal Asia. However, both MODIS datasets show more $AOD_{550} > 0.8$ days over India and portions of eastern Asia, and the MODIS

DT/DB captures more $AOD_{550} > 0.8$ days over the central Atlantic Ocean. Without having an exact truth to provide validation of $AOD_{550}$ over whole regions, it is difficult to determine which best captures severe events, although it is suspected that differences can come from several sources. For example, the differences over the central Atlantic Ocean between MODIS DT/DB and VIIRS may be due to the difference in the dust models used in the algorithm over ocean. This is reflected in the high bias in the MODIS DT/DB AOD over the N. Atlantic Ocean.

A second type of threshold score is slightly more relative, by calculating the probability of each dataset capturing a 95th percentile event compared to the total number of detected 95th percentile events detected by at least one other algorithm as well as the probability of only a single dataset detecting a 95th percentile event. That is, relatively speaking, we ask is a 95% from one set matched by 95% of another, thus accounting for slight biases in the datasets. Of course, if no dataset captures a 95th percentile event then it can't be counted. This metric is used to identify how

common it is for the datasets to be in agreement and identify areas where the detection of 95th percentile events is missed by the individual algorithms with the results presented in Fig. 4. In order to focus on high aerosol events, a threshold was set to eliminate points where the 95th percentile $AOD_{550}$ was less than 0.3. It is important to be reminded that this metric does not define the accuracy of capturing 95th percentile $AOD_{550}$ events, but rather the



consistency that two algorithms are in agreement that an event has taken place. Such consistency is required to
bridge the climate data record between sensors and algorithms and important for data assimilation.

Overall datasets are generally in agreement when detecting 95[th] percentile aerosol events, particularly in regions
where high aerosol loadings occur. Not surprisingly with its increased coverage, VIIRS DB shows the highest
likelihood of identifying an event at the 95[th] percentile. VIIRS DB is more likely to detect individual 95[th] percentile
events over ocean. Over land, MODIS MAIAC identifies more individual 95[th] percentile events than the other two
datasets.

MODIS DT/DB was regionally inconsistent in detecting 95[th] percentile events, especially over northern South
America, central Europe, northeastern Asia, and the central Atlantic Ocean. This may be due to the cloud
conservative nature of the algorithm in order to minimize cloud contamination. This may also be due to a sampling
related difference as MODIS DT/DB products are available at 10x10 km resolution at nadir while MODIS MAIAC
and VIIRS DB aerosol products are available at 1x1 and 6x6 km resolution at nadir, respectively.  MODIS DT/DB
and MODIS MAIAC show lower detection rates along the ocean coastlines of Asia with respect to VIIRS DB with
all three datasets seem to be detecting different events within central Africa.

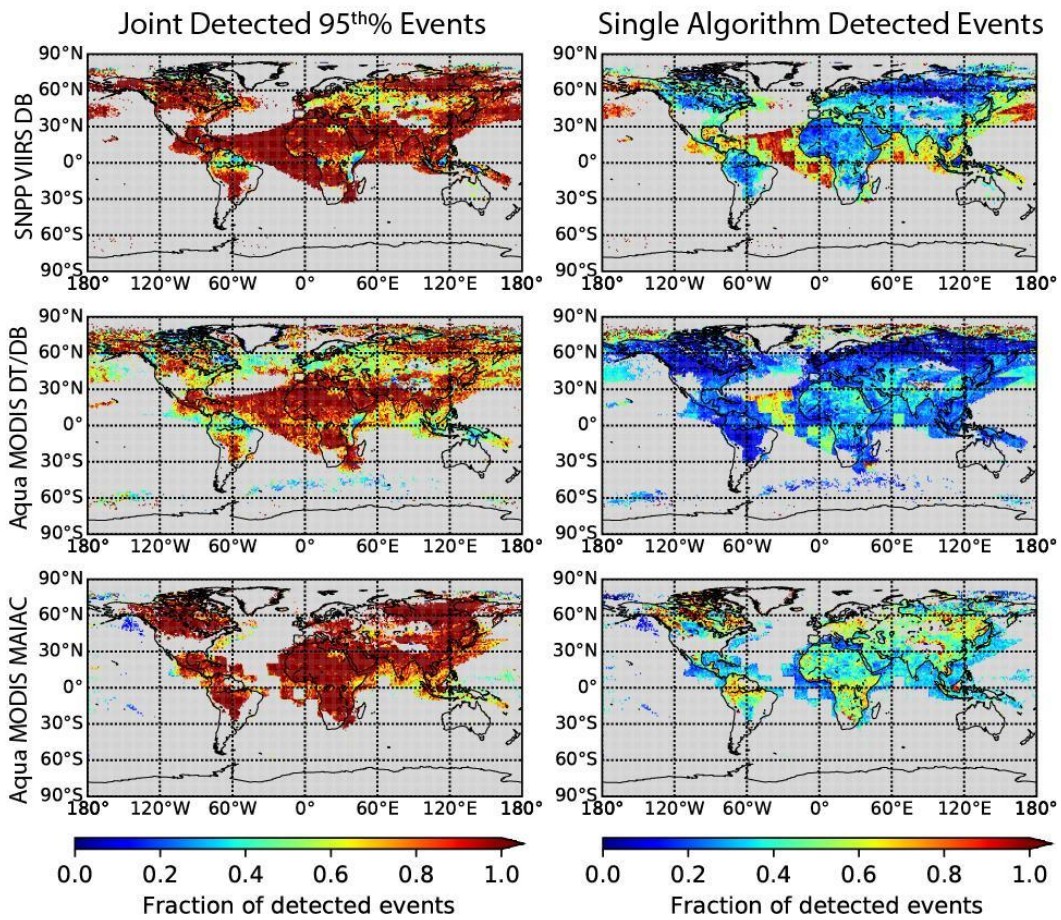

**Figure 4.** Consistency of detection for 95th percentile events detected by each dataset compared to the total detected 95th percentile events (left) and amount of 95th percentile events detected by a single sensor/algorithm in comparison to total detected 95th percentile events (right) for VIIRS DB (top), MODIS DT/DB (middle), and MODIS MAIAC (bottom) with a threshold for 95th percentiles > 0.3

### 3.2 Global Pairwise Analysis

While the comparison of the probability distributions of datasets characterizes overall sampling, we also wish to know how products compare at individual points and times, especially for significant AOD events. Here we briefly repeat the pairwise global analysis of Shi et al., (2011) with the updated algorithms and an emphasis on higher AOD regimes where we expect nonlinearities in AOD to exist between products. The global analysis presented here will then feed discussions of specific regional phenomenology that are provided in Section 4. While regression is a

useful tool, there are considerations when interpreting the results. The quality of a relationship is often indicated in the coefficient of determination ($r^2$) which provides the fraction of variance captured by a regression line. Thus, for a given error bar (say +/- 0.1 in AOD), data with wider dynamic range will by nature have a higher AOD, and low



AOD environments are penalized by the $r^2$ metric. Further, for higher AOD events, retrieval microphysical assumptions and degrees of freedom (absorption, size/refractive index/phase function etc.) should create a host of

local nonlinear and multi-modal relationships between products. Indeed, the products examined here do not even differentiate between fine and coarse mode over land other than what is regionally programmed. Errors due to lower boundary conditions should diminish with increasing AOD. Seasonal differences often exist. Ultimately, linear regression, while useful, is not a universal tool.

To account for nonlinearity, calculations were made two ways. First, linear regression is performed on the 1x1
degree bins for when $AOD_{550}<0.8$ for either of products being regressed (Fig. 5). Above this value, we found nonlinearities became prominent. Second, to extract AOD dependent biases we calculate pairwise mean bias between products for differing optical depth bands (Fig. 6), less than the median (<0.15), moderate AOD in the linear regime (0.15-0.4), transition to multiple scattering (0.4-0.8), multiple scattering (0.8-2); and exceptional $AOD_{550}$ (2-3). As MAIAC is limited to $AOD_{550}$ to 3, we ended the comparison there. But neither of the other
products generated a product for $AOD_{550}>4$, and even then, pairwise occurrences were exceptionally rare. As noted in Section 4, even this formulation is inadequate for regions with multi modal behavior.

As in Shi et al., (2011), strong spatially correlated biases are evident in Fig. 5 and 6, as evidenced by widely varying, slope, intercept and r2 values. Regions with the highest r2 are coastal waters, no doubt aided by the dark ocean boundary conditions and higher AOD relative to the open ocean. Slopes are also reasonably close to one for VIIRS
and DT/DB products, with DT/DB slightly low differences from 0-20% for moderate AOD. DT/DB also has a slight positive offset. MAIAC coastal waters show stronger gradients in slope to the traditional dark target counterparts, perhaps due to its use of more prescriptive optical models.

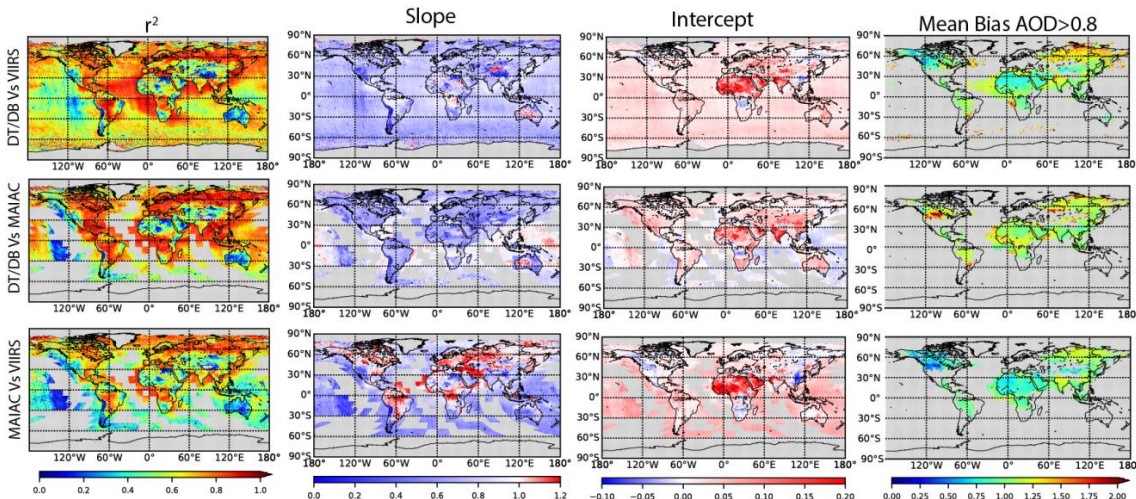

**Figure 5.** Pairwise regression and mean difference maps between the NASA VIIRS and MODIS Dark Target/Deep Blue (DT/DB) and MAIAC optical depth products. Regressions are performed for $AOD_{550}<0.8$, with mean relative bias between products from $AOD_{550}>0.8$.


Over land products continue to show significant spatial correlation of AOD between products. Over land, the highest
correlations are for moderate to high AOD regions over low-albedo vegetated lands including biomass burning
regions of South America, boreal Asia, central Africa, and peninsular Southeast Asia, as well as pollution dominated
regions of the eastern United States and Europe. In contrast, regions of low $r^2$ values, and hence indeterminate value
of slope and intercept, include the low-AOD areas of the tropical to subtropical Pacific Ocean and the deserts and
mountain ranges of the western United States, Chile, central Asia and Australia. However, arid areas with
moderately strong AOD signals also compare poorly, especially Saharan Africa and Southwest and Central Asia.
Large intercept deviations between products are generally highly localized, likely due to arid land surface features
(desert playa, rocks, bare soil), and orographically related haze and dust features (such as in India and western China
as discussed in Section 4). Saharan Africa, with its proclivity for dust production, shows the most dispersion
between products for a region of moderate to high AOD. Mean biases from high AOD events likewise have strong
regional patterns, again perhaps due to different dust models.

Like the probability distributions, regression and bias statistics are often markedly different across the land and ocean
boundary-especially between arid regions and water. Excellent examples of sharp gradients in model comparisons of
this include the transition from Saharan Africa to the Atlantic Ocean and south and southwest Asia to the Arabian
Gulf and Bay of Bengal. While it is surmised that this is largely due to lost signal to noise over bright backgrounds, it
is a reflection of the differences in over land and over water portions of retrievals. Finally, while mostly regressed out
in the pairwise comparison, one must also consider that the very sample populations are different on either side of the
shoreline. Influences include orography, glint, and differences in cloud features. The AOD distributions also may be
different due to land/sea breeze interactions. Over water, sun glint removes up to a third of the large fraction of the
swath. In all these cases, examples are provided in the regional results section, and even a cursory view of the NASA
worldview site (https://worldview.earthdata.nasa.gov/) will show numerous examples of retrievals being available of
only one side of the shoreline.

As noted earlier, while product to product regressions are quite useful, they do not explain why one sensor/algorithm
might have difficulty capturing AOD related dependencies to microphysics and lower boundary condition. By
looking at product differences by regime or region, one can begin to infer how algorithm assumptions are
influencing them. The largest relative differences are at the lowest AODs, well correlated with lower boundary
condition. For DT/DB-MODIS versus VIIRS, they are seen at the obvious geographic boundaries between dark and
bright surfaces, and in the case of MAIAC relative to traditional dark target, the land sea boundary. As AODs
increase, the gradients in AOD ratios between products decrease, in part due the diminishing influence of the lower
boundary condition, but also due to a reduction in coverage of areas with higher AODs. Nevertheless, regional
biases remain for high AODs environments. For AODs higher than the regression range (i.e, $0.8<AOD_{550}<2$;
$2<AOD_{550}<3$), biases over regions still span +/-40% over large regions, with perhaps the strongest gradients along
the African and Southwest Asian Coast, with VIIRS having higher $AODs_{550}$ over MODIS. Interestingly, mean
biases over the Siberian Boreal Region are opposite in sign to most of Boreal Canada. Also notable are strong biases
(greater than a factor of 2 or even 5) over the high latitude regimes, presumably due to cloud contamination. Likely
the same can also be said about ice over the arctic regions. At the individual pixel level for high AOD regions, there





is still a great deal of variability, most likely due individual sampling differences between products along plume edges. All of these are discussed by region in the following section.

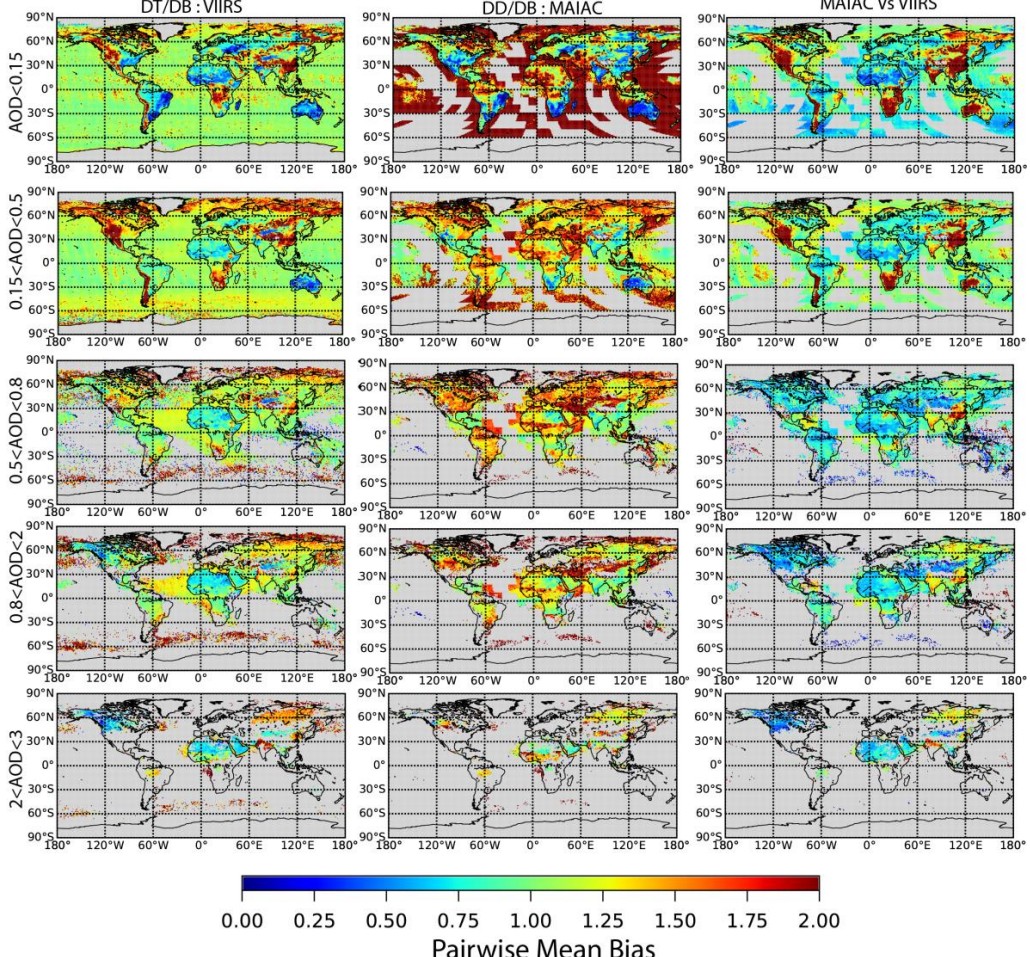

**Figure 6.** Pairwise mean biases between products for differing AOD ranges where at least one of the products has a value within that range.

## 4 Regional Analysis

The global overview provided an initial review of the relative distributions of high $AOD_{550}$ events between the three AOD products. But dissimilarities in the $AOD_{550}$ distributions can be a result of several aforementioned root causes of sampling, microphysics, lower boundary condition, and aerosol/cloud discrimination. To attribute product differences by region, nine regions as shown in Fig. 1 were selected for further discussion based on areas of higher geometric standard deviations, noticeable differences between the datasets, and variation in fractional identification

of high percentile occurring events. Each region demonstrates its own challenges for retrievals and sampling.



To capture the differences in nature between the different regions, domains are intercompared in three figures. Figure 7 provides an area averaged time series of Aqua Dark Target/Deep Blue, Aqua MAIAC, SNPP-VIIRS and AERONET over the 2016-2019 study period. Figure 8 provides the corresponding scatter plots of the MODIS products and AERONET to the VIIRS product. Finally Fig. 9 plots log-probability of individual 1x1 degree point

that make up the regional averages used in Fig. 7 and 8. Even though AERONET's point nature results in limited coverage, its data are included in these plots to investigate local representativeness (as indicated in Fig. 1(a)). In addition to these composite figures, scatter plots between 1x1 degree products for each region are provided in supplemental material S1. In the following section, it examines biomass burning (e.g., Fig. 7 (a)-(e)), dust (e.g., Fig. 7 (f)), and pollution/mixed dominated environments (Fig. 7 (g)-(i)).



**Figure 7.** Time series of daily mean AOD from 2016-2018 for the regions presented in Fig. 1. Retrievals include AERONET (orange), VIIRS DB (blue), MODIS DT/DB (red), and MODIS MAIAC (green). Total 84th percentile (dot), 95th percentile (dot dash) and 98th percentile (dash) are also indicated.






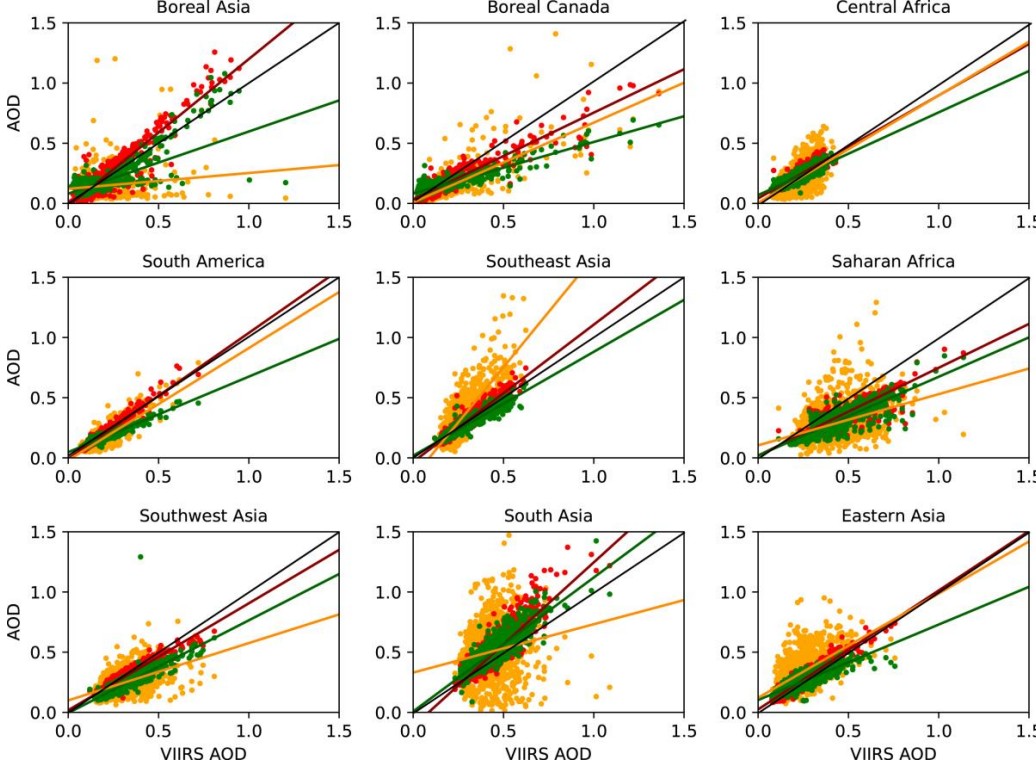

**Figure 8.** Regional scatter plots of time series data presented in Fig. 7. Retrievals include AERONET (orange), MODIS DT/DB (red), and MODIS MAIAC (green).Also shown is the 1:1 line (black).

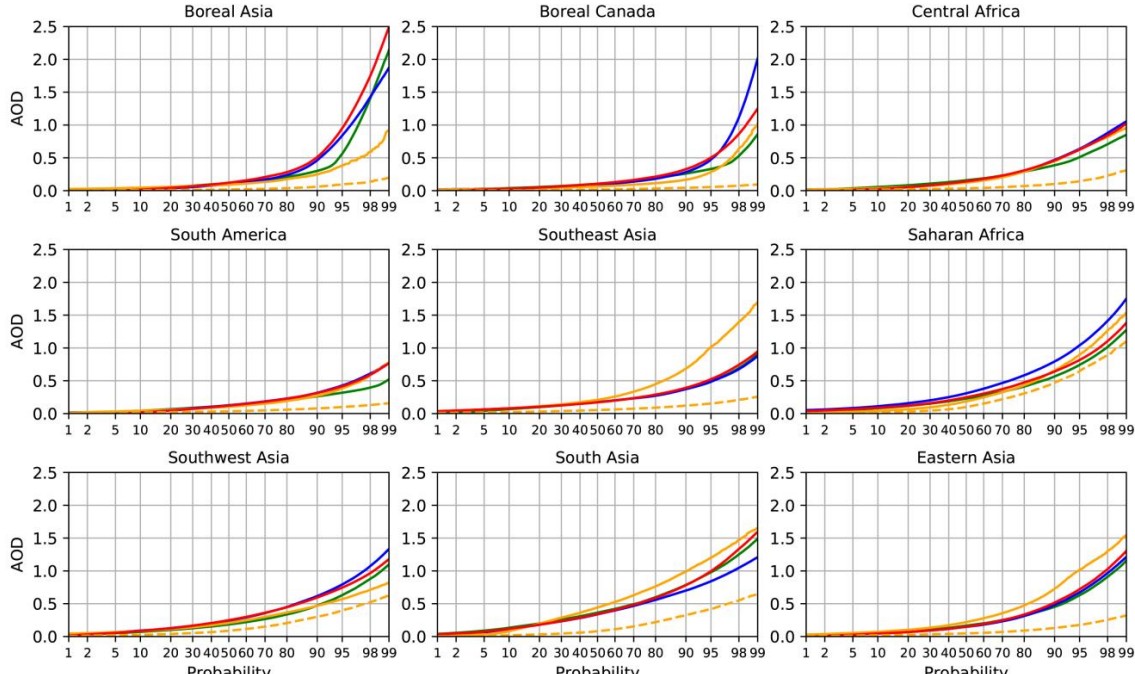

**Figure 9.** Log probability plots of 550 nm AOD taken from the 1x1 degree samples within the regions defined in Fig. 1. Retrievals include AERONET (orange), VIIRS DB (blue), MODIS DT/DB (red), and MODIS MAIAC (green). SDA derived coarse mode AOD from AERONET is also provided (orange-dashed)

### 4.1 Biomass burning dominated regimes

While Fig. 2 and 3 demonstrates that biomass burning is second to dust for median or even 84% $AOD_{550}$, any picture of a significant biomass burning event shows plumes have some of the very highest AODs on the planet, locally rivaling clouds. Even regionally, smoke AODs from fire complexes can be extreme. Peat burning in Indonesia can generate $AOD_{550}$ values that can only be estimated by AERONET's near infrared wavelengths, indicating mid visible AOD>10 (Eck et al., 2019, Shi et al., 2019). Likewise AERONET data corroborate dramatic photographs of the San Francisco Bay Area during the Sept 2020 fire season (outside the period of study) led to twilight conditions at solar noon (https://theglobalherald.com/news/wildfires-on-us-west-coast-turn-day-into-night-dw-news/; last accessed 10 AUG 2021). It is this extreme behavior of smoke emission that makes the quantitative monitoring of biomass burning by satellite so challenging.

#### 4.1.1 Boreal Regions: Continental and Intercontinental Scales

Boreal Asia and Canada (Fig. 7/8/9(a)&(b), respectively) are excellent examples of regions exhibiting extremely dispersive $AOD_{550}$ distributions, with low median values but occasional severe continental to intercontinental scale wildfire smoke events. As shown below, this aerosol domain is perhaps the most difficult to assesses. Nevertheless,





within the regional spatial average domains, all three satellite algorithms tracked one another well, even with very
sharp peak days without the gradual seasonality seen in other regions (Fig. 8(a)&(b)). With only 3 AERONET sites
in Boreal Asia, there are periodic spikes due to the proximity of fires to instruments. In comparison, the higher
number of AERONET sites in North America shows more convergence between satellite and sun photometers.
From a probability distribution point of view (Fig. 9), Boreal Asia clearly has the highest prevalence of detected

high AOD events globally, ranging from 1.7-2.5 at the 99% level, with Boreal North America showing more
dispersion between products with 99% AODs ranging from 0.8-2. Not surprisingly, AERONET demonstrates high
AODs are almost always associated with the fine mode. This said, high latitude dust does exist from isolated sources
or transport (Bullard et al., 2016). However, such events are unlikely to be detected by isolated AERONET sites or
identified as coarse mode dominated by the satellite products examined here.

Figure 7 and 8 statistics are consistent with Fig. 6; pairwise biases between products switch in sign between the
Asian and North American domain. Indeed, Fig. 9 and Fig. S1&S2 show that DT/DB and MAIAC provide 20%
lower values than VIIRS for the Boreal Asia. However, for Boreal Canada, the MODIS values are greater than
VIIRS by more than 50%. However, for these two regions, the differences between the satellite products are much
less different than the differences between the satellite products and the AERONET data used for verification. Given

there are so few AERONET verification points for high AOD and multiple populations visible between copious
satellite data populations, one cannot say definitively which product is most "correct." Assumptions on an individual
retrieval's overland size /optical properties (most notably absorption as AOD increases) coupled with multiple
scattering are expected to result in spatially correlated differences between products. Indeed, mid visible single
scattering albedo can vary considerably by individual fire and age, ranging from 0.9-0.99 (Reid et al., 2005; Eck et

al., 2009; Nikonovas et al., 2015).
The extreme behavior of fires in boreal and temperate forested domains provide cautionary examples of sampling
and regional averaging. For this study, the most significant boreal smoke outbreak was observed over the North
American boreal region domain in early July 2018 (Fig. 7(b)), for which all products reported smoke AODs over 1.
Meanwhile Boreal Asia showed modest fire activity with AODs<0.5. However, this North American event was

Siberian in origin. Significant thermal hotspot anomalies and smoke build up over the Siberian Boreal started in
earnest was detected starting 26 June, 2018 in association with convection and consistent with lightning. By July 1,
AOD were well above 1 along a 2000 km band over Siberia (Fig. 10 (a)). By July 5, smoke was advected
northeastward into the Polar Regions along with significant cloud cover masking it from quantification.
This smoke outbreak was collocated with significant cloudiness and the retrieved AODs are likely biased high.

Much of the smoke plume that is cloud free has failed retrievals Fig. 10(b). By 9 July, the smoke is adverted
southeastward into Canada, again with relatively few observations relative to the 2000 km size of the plume Fig. 10
(c). Thus, these aerosol events, perhaps the largest by AOD and size, are a significant challenge to track and
apportion by standard satellite products alone.

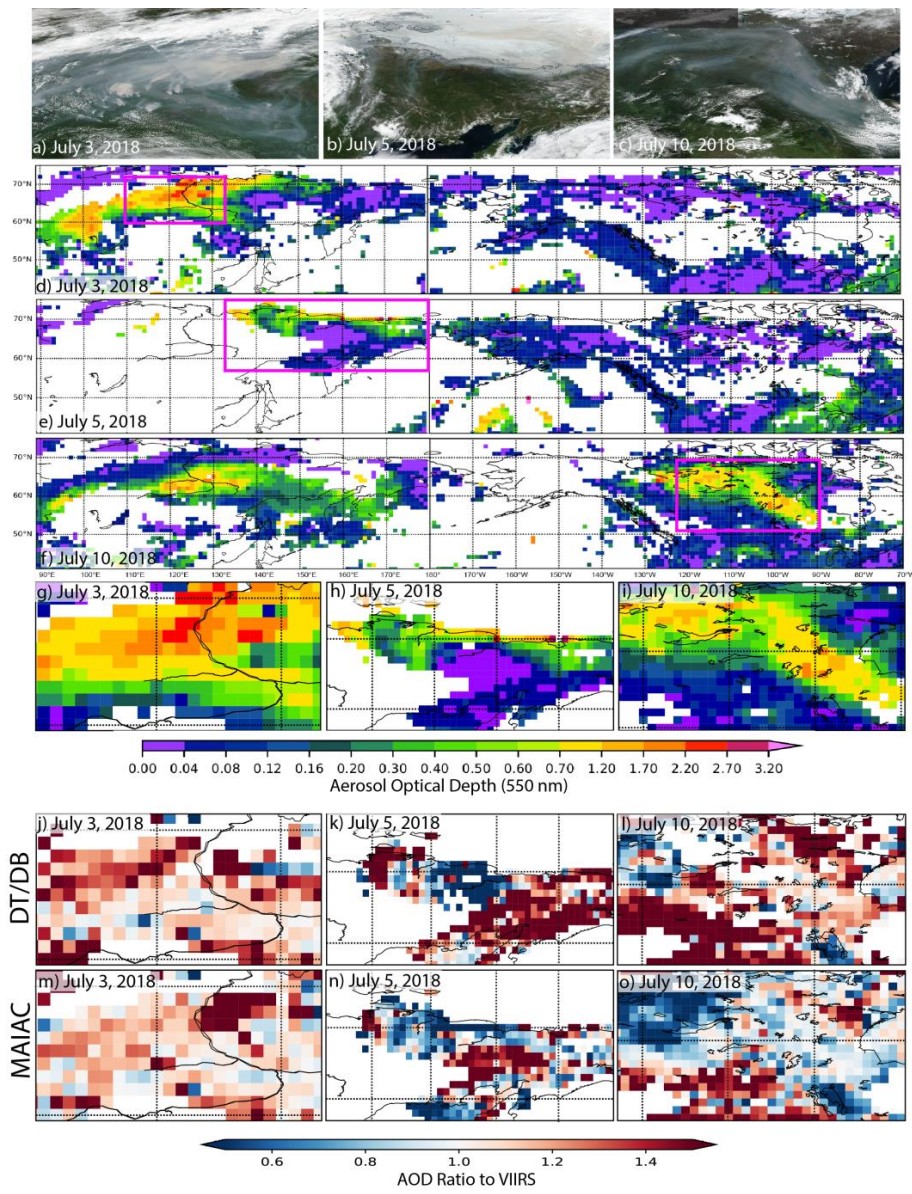

**Figure 10.** A case of Siberian smoke transport from Asia to North America for early 2018. a)-c) corresponding RGB images of plume evolution for July 3, 5, and 10, 2018, respectively. d)-f), corresponding VIIRS AOD550 for Asia through North America. The magenta box corresponds to the images in (a)-(c), zoomed in (g)-(i). (j)-(l) and (m)-(o) ratios of DT/DB and MAIAC to VIIRS, respectively. Satellite RGB imagery is from NASA Worldview.




A second difficulty is that, for Boreal Asia, significant AOD events are not just biomass burning. As a second example, strong regional $AOD_{550}$ were logged 6 March 2016 (Fig. 11(a)) for both DT/DB and VIIRS, appearing to be the fringes of a dust storm. Hand inspection of these cases were nearly entirely covered by snow and clouds (Fig. 11(a)). However, there were just a few retrievals solely associated with a significant Asian dust event along the

southern edge of the domain dominating the area average. Such events are exceptional, but obviously not impossible. Given that imager retrievals lack information content for fine-coarse partition, aerosol sources can easily be confused. Indeed, numerous thermal anomalies are observed in the region (Fig. 11(a)) although they are persistent and likely from petrochemical flaring enhanced by the cold background. Such differences between products may become more significant as developers create different cloud and snow screening techniques. In light

of what is presented here, the observed integer factors in regional bias between products due to sampling and interpretation alone is not unreasonable, although the exact source of the error will require further study.

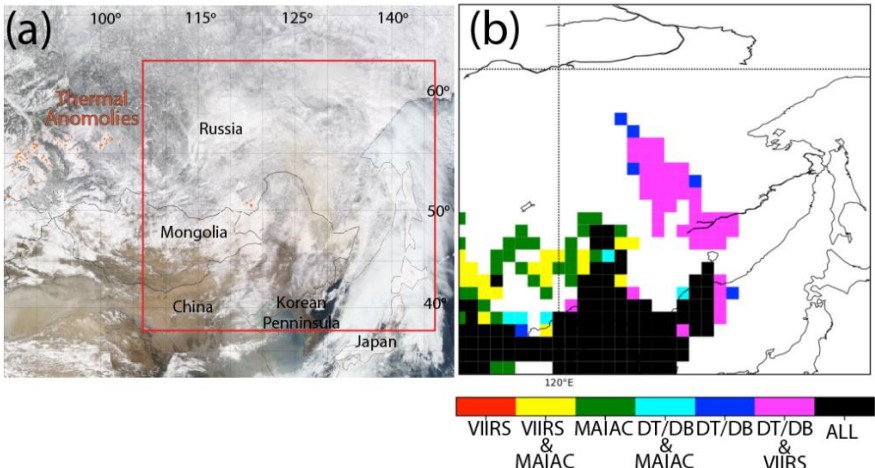

**Figure 11.** Case study of a springtime dust intrusion from China into the Siberian domain on 6 March, 2016. Included is (a) RGB image of dust transport and MODIS thermal anomalies; and (b) a map of coverage of where retrievals are available. Satellite RGB imagery is from NASA Worldview.

### 4.1.2 Central Africa and South America: Variability within large scale plumes

In comparison to the extreme behavior of boreal biomass burning regions, the tropical to subtropical Central Africa

and South America burning regions have more manageable seasonal biomass-burning signals, dominated by the August to October period (Fig. 7 (c) and (d), respectively). Instead of isolated mega fires with exceptional transport phenomenon, smoke is generated from numerous small grass and deforestation burns that merge into regional plumes embedded in more subdued meteorological regime (Reid et al., 2009). These regions also have significantly higher AERONET site prevalence compared to the boreal region, allowing for more assessment opportunities (Fig.

1, Fig. 8, S3&4). Over these domains, satellite products and AERONET track well (Fig. 7), and with the exception a



slight discrepancy for MAIAC probability distributions, overlay each other almost exactly (Fig. 9). Over Central Africa satellite products generally show low scatter between each other, although in pairwise comparisons to AERONET they all show a distinct low bias (Fig. S3). Being the world's largest biomass burning source (Mu et al., 2011), Central Africa has a smaller range of regional optical depth than other biomass burning regions. South America in contrast, has even better pairwise constancy between products (Fig. S4) but MAIAC and VIIRS have lower values relative to DT/DB.


While bulk comparisons suggest overall agreement between products, it is concerning that pairwise model biases do not always manifest themselves in comparisons to the prevalent AERONET sites in the region despite these region's broad plume features. Further, regionally prescribed $\omega_o$ and differences in over water retrievals result in sharp discontinuities in inter-product bias (Fig. 5 and 6). For Africa, a well-established seasonal cycle in $\omega_o$ starting at ~0.84 in the early season due to a high prevalence of grass fires and increasing in time to 0.93 with increased forest fuels until the end of the season manifests itself in a seasonal cycle in MODIS bias (Eck et al., 2013).


Figure 12 provides plots of Sept 9, 2018 as an example day of interproduct differences, including (a) an Aqua MODIS RGB; (b) coverage diagram; (c), (d), (e) DT/DB, MAIAC, and VIIRS AOD, respectively, with AERONET AODs overlaid; (f),(g), (h), interproduct ratios of DT/DB to VIIRS, MAIAC to VIIRS, and MAIAC to DT/DB, respectively. This example was picked as a typical day, but with an Aqua orbit gap in the middle to allow for examination of extreme differences in scattering angles. At first glance, AOD patterns do match well overall, but there are some spatially correlated regions of difference. These are highlighted in the ratio plots. Notable are differences include 1) magnitude, with the largest ratios being associated not with higher but lower AODs; 2) All three products compare well to AERONET on the eastern portion, but low bias in the west, possibly due to differences in smoke optical properties (worthy of future study). The next version of VIIRS V2.0 has been described as improved by its developers; 3) there is a sharp coastal delineation in AODs across the AOD values matching the near shore AERONET sites. 4) DT/DB relative to VIIRS has a high bias of up to 30% along the edge of swath views; 5) DT/DB also has varying biases relative to VIIRS offshore of the eastern coast, with higher in the north with low AOD, a good match west of Madagascar for moderate AODs, and a slight high bias south of Madagascar. MAIAC yields higher values than the others to the north, and lower to the south. 6) MAIAC has rectangular regions of bias that are quite distinct from both DT/DB and VIIRS. This is likely due to the tile processing nature of MAIAC.




Resolving all these issues is far outside the scope of this study. But the conclusion here is that it is clear that on a daily basis that significant spatially correlated biases exist between products based on a host of sources even if on a larger scale they converge to similar average values. This finding requires consideration when the data products are used for high fidelity data assimilation and inverse modeling applications.




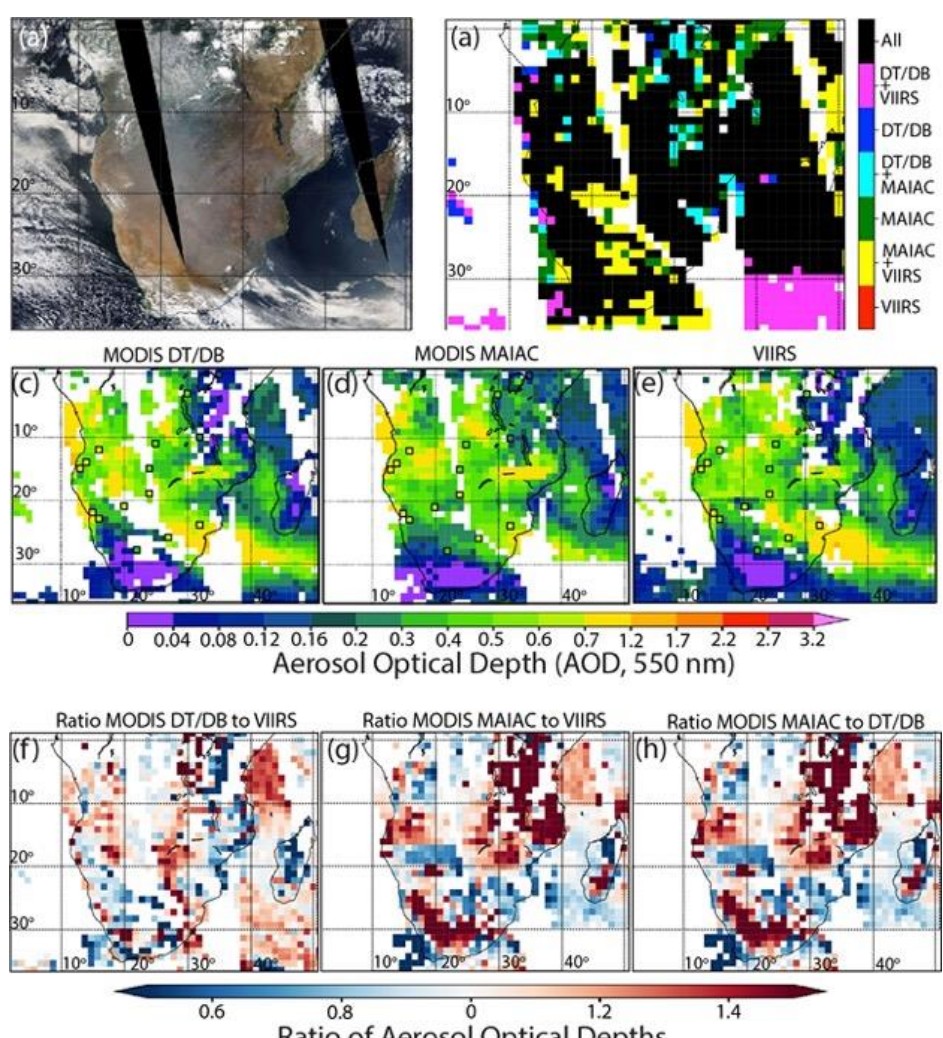

**Figure 12.** Intercomparison of aerosol retrievals for Southern Africa for Sept 19, 2018. (a) Aqua RGB image; (b) Locations where retrievals are available; (c)-(e), MODIS DT/DB, MODIS MAIC and VIIRS AOD$_{550}$ retrievals, with AERONET AOD$_{550}$s in boxes. (f)-(h) ratios of retrieved AOD$_{550}$ to each source. Satellite RGB imagery is from NASA Worldview.

### 4.1.3 Southeast Asia: Exceptional AODs

Southeast Asia has two biomass burning seasons: 1) boreal spring agriculture, deforestation, and wildfires in mainland Southeast Asia countries of Myanmar, Thailand, Cambodia and Laos; and 2) boreal summer and early fall agriculture, deforestation and peat fires for Maritime Southeast Asian countries of Indonesia and Malaysia (Reid et al., 2012). Mainland Southeast Asia tends to have more consistent seasonal behavior with periodic enhancements





(Reid et al., 2013), similar to the Africa and South America regions. Maritime Southeast Asia, however, has strong variations due to El Nino-Southern Oscillation (ENSO; Nichols, 1998; Field et al., 2016) as well as a host of interseasonal meteorological dependencies such as the Madden Julian Oscillation/ and Boreal Summer Intraseasonal Oscillation (Reid et al., 2012). AODs can be exceptionally high for weeks at a time, so that even for wavelengths as long as 870 nm, sun photometers have insufficient solar signal (Eck et al., 2018). Given its high regional cloud cover, extremes in AOD, and variable aerosol optical properties, the Maritime Southeast Asia is exceptionally poor at sampling these extreme conditions (Reid et al., 2013). Yet, all three satellite products track each other exceptionally well at characterizing regional average (Fig. 7&8) and show similar probability functions (Fig. 9). Even scatter plots between products show good comparisons (Fig. S5).

Despite the excellent overall comparability, satellite products under-sample extreme events observed by AERONET. Indeed, 99th percentile AOD for the satellite products are only 60% of the AERONET values (0.95 versus 1.6). This behavior has already been well-documented by Reid et al., (2013) and Eck et al., (2018) for El Nino induced drought periods. Now with Version 3 of AERONET, it can infer $AOD_{550}>5$ by inferring from the 1020 nm channel. For example, the year 2019 in our study period had exceptional biomass burning AOD. Figure 13 provides a timeseries of 550 nm AOD from AERONET at three sites on Borneo: (a) Palangkaraya (2.3 S; 113.9E), in the heart of burning activity in southern Kalimantan Indonesia with maximum measured AODs; (b) Pontianak (0.1N; 109.2 E), on the shore of western Kalimantan as smoke exits to Java Sea; and (c) Kuching (1.5 N; 110.3 E) on the southern border of Sarawak, Malaysia as smoke exits into the South China Sea). Over southern Kalimantan in the heart of the source region, we find the greatest discrepancies between products. For the whole month of August, AODs from the satellite products significantly underestimate smoke levels. By mid-September, when smoke loadings are at a maximum, the sun photometer saturates, and AODs over 4 can only be estimated by using near-infrared wavelengths. Given saturation, estimates shown in this figure are not even included in the probability distributions shown earlier in this paper. Owing to its higher resolution, MAIAC has additional coverage than the other products. By the end of the season when AODs diminish, AERONET and satellite products reconcile. Along the coast at Pontianak, products compare well between themselves and AERONET, we hypothesize due in part to smoke homogenization from many plumes into a single high AOD region. However, the highest AODs are still understandably missed. By Kuching, smoke AODs are lower still, and all products compare well.

An example day is provided in Fig. 14, in a manner similar to Fig. 12 for Central Africa, with RGB, coverage, AOD and ratio plot panels. The difference between this case and Africa is striking. Whereas Africa showed good coverage across all products, Borneo shows slight changes in AOD and cloud mask thresholds as well as increased VIIRS coverage which results in more variable retrieval coverage. All products miss the center of Kalimantan, due to extreme aerosol conditions. Along the border of the plume, ratios between products can be extreme depending on the individual retrievals that make up the aggregates. The conclusion in this case is that all retrievals have some physical limits. To cope when AODs are this exceptional, new techniques need to be developed for measurement, aggregation and assimilation.





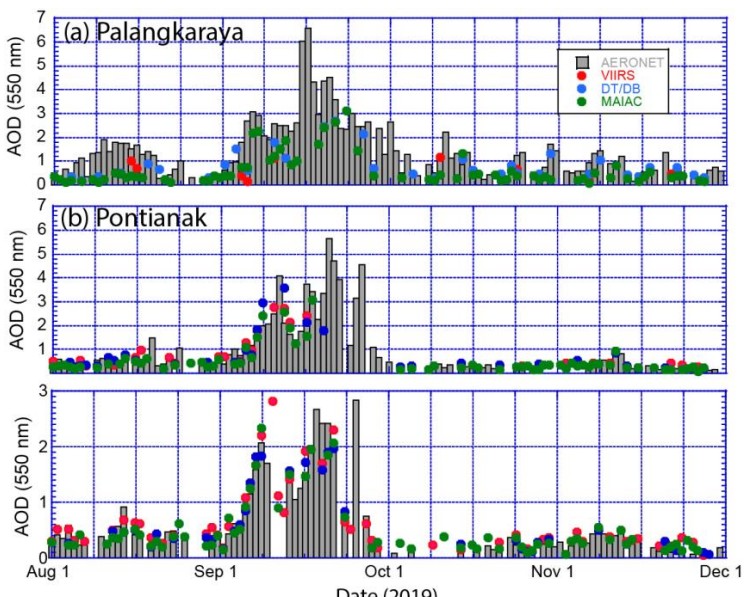

**Figure 13.** Time series of satellite product AOD to AERONET for 3 sites on Borneo for the 2019 burning season (a) Palangkaraya, southern Kalimantan, Indonesia; (b) Pontianak, western Kalimantan, Indonesian; (c) Kuching southern Sarawak, Malaysia.



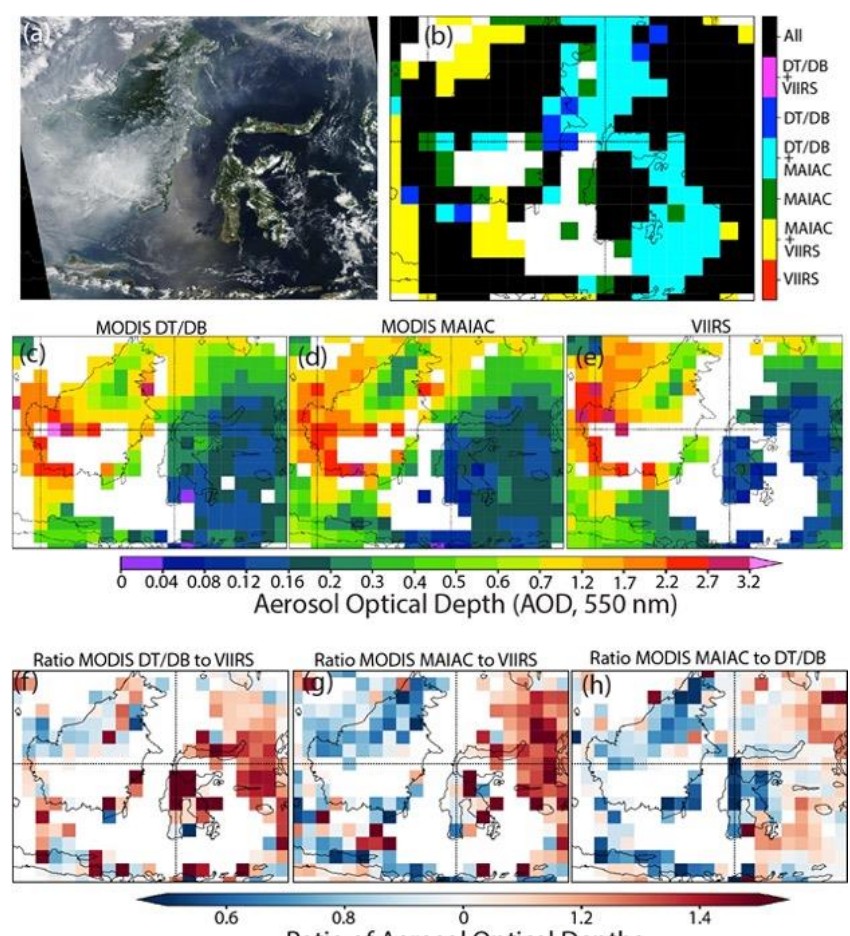

**Figure 14.** A Borneo image and AOD analysis for Sept 16, 2019. Included is (a) corresponding Aqua MODIS RGB image; (b) coverage coincidence map (c) MODIS DT/DB AOD; (d) MODIS MAIAC AOD; and (e) VIIRS AOD. Also shown are the ratios between products, including (f) Aqua DT/DB to VIIRS; (g) Aqua MAIAC:VIIRS; and (h) Aqua MAIAC: MODIS. Satellite RGB imagery is from NASA Worldview.


**4.2 Dust Dominated Saharan Domain: Bright surfaces and dust microphysics.**

The North African/Saharan region is the only subcontinental domain that can be said to be fully dominated by dust. With median AODs on the order of ~0.3 and 98% values of 1 to 1.5 by product, the Sahara is the largest contiguous aerosol feature on earth. Visible in Fig. 7 is the Sahara's dynamic nature with frequent region-wide spikes in

$AOD_{550}$. While the dust season is often envisioned as comprised of massive boreal summer Saharan Air Layer



outbreaks traversing across the subtropical Atlantic into the Americas, major events can occur any time of year with only a minor boreal winter minimum.

Generally, regional dust products are comparable for regional average (Fig. 7, 8) and probability density (Fig. 9). Even from a bulk point-by-point comparison (Fig. S6), the products correlate well to themselves and with the few

AERONET sites in the region. VIIRS DB tends to be consistently higher than MODIS counterparts by up to 30-50%, most likely due to assumptions in dust optical properties as well as perhaps due to some improvements in cloud/dust discrimination. In contrast, MODIS MAIAC shows the lowest daily average of $AOD_{550}$, which is a difference that persists throughout the analysis of severe events. The AERONET values are less correlated with the satellite observations, likely a result of the sparse sampling in this region. However, close examination of these plots

(most notably S6) in combination of Fig. 5 shows there are multiple data populations embedded into the whole with regions of significant decorrelation and bias. Areas of particular decorrelation are coincident with areas that are dominated by evaporates (i.e., low Fe absorption), such as the Tenere Desert/Bodele Depression of Chad, Qattara Depression of Libya and Egypt and the Western Sahara of Mali, Mauritania, Morocco (Goudie et al., 2002; Perlwitz et al., 2015).

The spatially correlated nature of bias between products is provided as an example in Fig. 15 for June 6, 2017. Given deserts' bright surfaces, dust may not be visually obvious (Fig. 15(a)). However, there is minimal cloud cover and excellent satellite coverage is available (Fig. 15(b)). All three products show the same overall dust features, of multiple dust plumes (Fig. 15(c)-(e)). However, by ratio, significant patterns emerge (Fig. 15(f)-(h)), sometimes with opposite signs in bias for adjacent plume features (e.g., western Africa), along straight lines due to regional

boxes used in the retrieval, along coastlines, and on either side of an orbit where scattering angles abrupt change. Of all the cases shown in Section 4, Fig. 15 best demonstrates the challenges of assimilating or performing source function inversions. Because data assimilation must account for observation localization and there are such few temporal observation opportunities to begin with, differences such as these result in a smeared source area (Khade et al., 2012). Further, since there are so few AERONET sites available and day to day changes in solar geometries,

these differences are difficult to deconvolve.



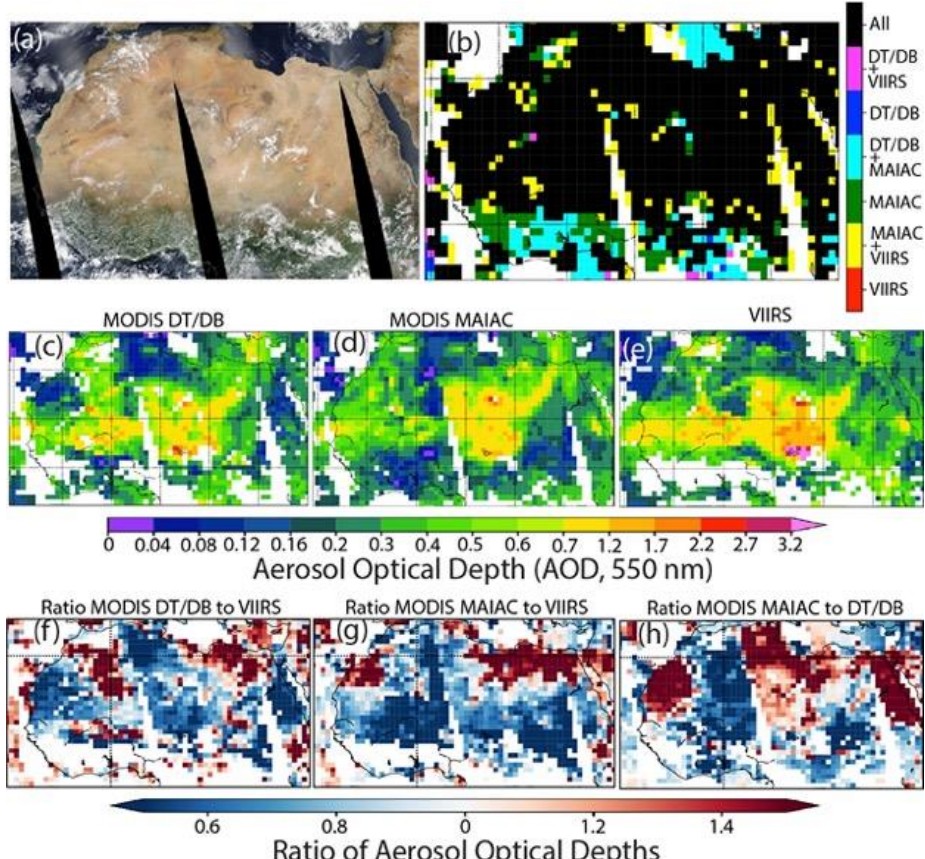

**Figure 15.** A pan Saharan image and AOD analysis for June 8, 2017. Included is (a) corresponding Aqua MODIS RGB image; (b) coverage coincidence map (c) MODIS DT/DB AOD; (d) MODIS MAIAC AOD; and (e) VIIRS AOD. Also shown are the ratios between products, including (f) Aqua DT/DB to VIIRS; (g) Aqua MAIAC:VIIRS; and (h) Aqua MAIAC: MODIS. Satellite RGB imagery is from NASA Worldview.

### 4.3 Mixed pollution/dust domains of Asia

It so happens that some of the regions with the strongest pollution emissions are also influenced by dust transport. Indeed, the coastal arc extending from the Arabian Sea, through India and up to East Asia hosts some of the most heterogeneous "mixed" aerosol environments of the world. Figure 9 shows that Southwest Asia and southern Asia have coarse mode AODs on the order of 50% of the total value out past the 99th percentile. East Asia, known for its significant haze dominated by the fine mode, nevertheless is frequently impacted by dust storms from central Asia, such as the Taklimakan and Gobi deserts. Like the Sahara, correlations and biases across the SW to East Asia arc have strong spatial variability (Fig. 5, 6). Correlations are best for northern mainland SE Asia due to having dark,






vegetated surfaces and sufficiently large biomass burning sources. Areas with the lowest agreement include bright deserts, especially areas with aerosols having low values of light absorption.

In Asia, there is so much aerosol activity that numerous individual events can be observed on most days. Figure 16 provides a comparison for November 3, 2018, which includes significant dust events over SW Asia, biomass burning and pollution over India, and haze over western China (Fig. 16(a)). Like other cases, products compare well

qualitatively (Fig. 16(b)-(d)) but there are regional differences over both land and ocean (Fig. 16(e)-(g)). Each region has its own characteristics that are described below.

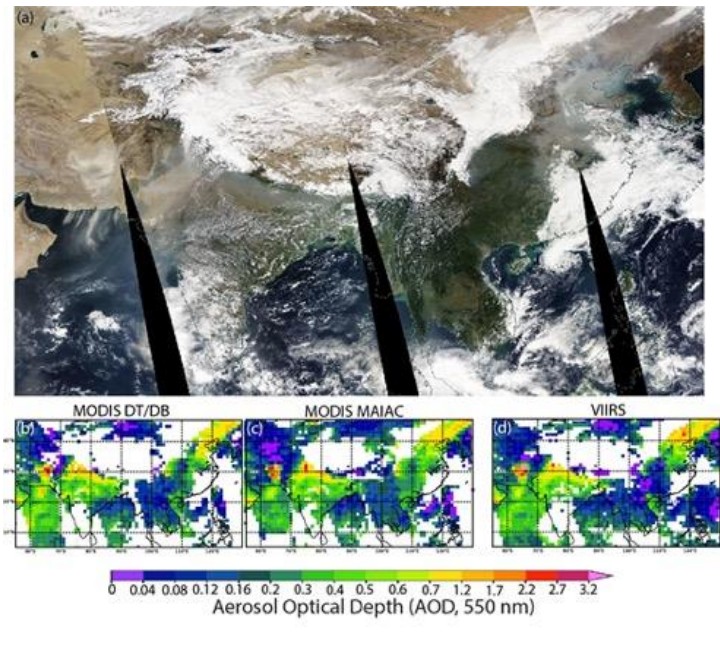

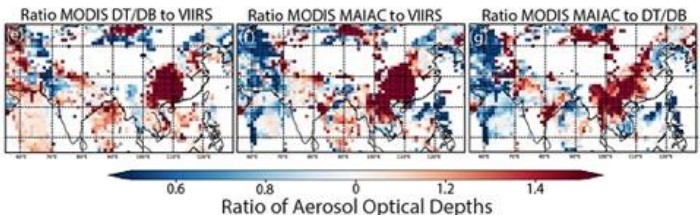

**Figure 16.** A pan Asian image and AOD analysis for Nov 3, 2018. Included is (a) corresponding Aqua MODIS RGB image (b) MODIS DT/DB AOD; (c) MODIS MAIAC AOD; and (d) VIIRS AOD. Also shown are the ratios between products, including (e) Aqua DT/DB to VIIRS; (f) Aqua MAIAC:VIIRS; and (g) Aqua MAIAC: MODIS. Satellite RGB imagery is from NASA Worldview.

### 4.3.1 Southwest Asia





Southwest Asia is a significant producer of dust, although less than the Sahara. Like the Sahara, biases are expected

due to lower boundary condition, dust microphysics, and optical geometry. Taken as a whole, products do track each

other reasonably well at the seasonal level (Fig. 7, and 8) and even look reasonable in the context of a point-by-point

scatterplot (Fig. S7), albeit with some biases. Imbedded in these regressions and many populations, and regression

between products at the 1° level are likewise poor over land, but good over water (Fig. 5). AOD dependent biases

exist between products, especially along the land-ocean border. However, complicating matters beyond the Sahara is

that owing to its large petrochemical based economy Southwest Asia also can exhibit exceptionally strong pollution

events (e.g., Smirnov et al., 2002; Reid et al., 2008) as exhibited in half the 84% AERONET AOD being fine mode

(Fig. 9 (g)). Thus, while the very largest events are dust dominated, it is not necessarily a given that moderately high

AOD is dust-potentially leading to confusion for overland algorithms that cannot extract fine/coarse partition.


### 4.3.2 South Asia

Not surprisingly, the South Asian/Indian Subcontinent domain, with its diverse sources from pollution, dust

transport, and agricultural burning, has the highest median AOD of the regions studied here. Like almost all regions,

products generally agree on the overall distribution statistics (Fig. 8(h)). Although there are many AERONET sites

in the region, AERONET averages depart sharply from the satellite products. Examining the time series further (Fig.

7(h)), the dominant aerosol features are biomass burning during the fall, and haze during the winter and early spring.

Pre-monsoon can also be associated with significant coarse mode dust. However, taken at the subcontinent scale, the

largest AODs detected by the satellites occur during the northern hemisphere summers, and the MODIS DT/DB

retrieves the largest. Yet, AERONET retrievals are also not well correlated with the regional satellite depiction,

showing higher AODs overall, but lower values in the monsoon period. This difference between satellite and

AERONET are likely resulting from a) the limited network in this region, notably concentrated in the Indo-Gangetic

plain; and b) cloud contamination. To investigate these large disagreements, case studies where selected using

NASA WorldView where it shows that there is a correlation between occurrences of peak $AOD_{550}$ in the satellite

datasets and low amounts of $AOD_{550}$ from the AERONET sites. These correlations arise when there are high AOD

events in the northern portion of the country along the Indo-Gangetic Plain of India. In this region, the AERONET

stations are located at boundary of the regions with high amounts of AOD. This similarly occurs for the opposite

case where the satellite datasets observe lower amounts of AOD and AERONET observes large amounts of aerosol

and helps confirm that sampling bias is causing the disagreement in the regional analysis.

### 4.3.3 Eastern Asia



Completing the coastal arc to east Asia, this region shows better inter-product agreement than any other in this category. This is despite being one of the most heterogeneous environments of the world, with significant dust transported from the Taklimakan and Gobi deserts, winter-time haze over the east China Plain, severe pollution from the Pearl River Delta and industrial centers, and biomass burning intrusions from Southeast and boreal Asia. There are also sharp gradients in land surface properties. Similar to other arid regions, Fig. 5 shows that far western China

has the poorest relationships between products. There is gradual improvement eastward. But, taken as a regional average, products largely converge-with the exception of high biases of AERONET sites- a result of sites being selected to monitor some of the most polluted areas of an already highly polluted region. Indeed, even though the region is known for all types of aerosols, AERONET is generally situated in populated areas, likely more reflective of pollution sources with high fine mode fractions.


**5 Discussion and Conclusions**

It is expected that in late 2023 there will be a marked shift in global aerosol monitoring as the 20+ year MODIS instruments are decommissioned, and the community must finish its adaptation to SNPP/JPSS VIIRS instruments. This transition will have a notable impact on such applications as aerosol data assimilation and the generation of

consistent climate data records. An area of particulate concern is in the monitoring of extreme events that already stress aerosol algorithms and are expected to become "more extreme" in frequency and magnitude with ongoing climate change. Therefore, to examine potential differences in the efficacy of MODIS and VIIRS based AOD algorithms and what a 2023 change in sensor platforms will result in, severe aerosol events from VIIRS DB, MODIS DT/DB, and MODIS MAIAC are assessed at global, regional, and ground based sensor perspective from

730  2016-2019.

Using a consistent gridding methodology across products statistics of AOD by each product were generated to identify where most significant aerosol events have occurred, what is considered a locally exceptional event by region, and how these differences spatially vary between datasets. These findings include:

a)  The median AOD values show relative agreement between all three datasets. Thus, a dramatic shift in

typical AOD values as systems progress from MODIS to VIIRS is not expected. However, there are slight regionally correlated biases by region. VIIRS has a slightly higher bias in comparison to MODIS DT/DB and MAIAC in high aerosol producing regions. The largest median differences are seen in the clean regions such as the western United States, Gobi desert, and the Arctic with the MODIS products being higher than VIIRS. *Thus, pristine regions may show a decrease in AOD from MODIS to VIIRS.*

b)  When observing the 84th, 95th and 98th percentile of AOD values, the biases between products start to become ever more apparent by region, especially surrounding areas impacted by biomass burning. For example, for Africa, mid latitude fires, and the Boreal regions, MODIS DT/TB has consistently higher AOD values than VIIRS, with a more neutral bias for South America and a reversal in bias in sub-Sahel/tropical Africa. Conversely, VIIRS provides higher 84th% values for African, SW Asian, and Asian

desert regions. In regions such as the Arctic Ocean, Africa, and southwest Asia, the ratios of MODIS to VIIRS along the land/ocean border show a clear distinction between when land and ocean retrievals





algorithms are being used. MAIAC generally retrieves lower high percentile AOD values than the other retrievals. Areas of correlated bias exist between lower MODIS MAIAC values and the MODIS DT/BB and VIIRS counterparts. This reveals algorithm differences caused by aerosol speciation, single scattering albedo, and surface differences between land and ocean. *Thus, AOD observations for higher AOD environments will notably shift with the MODIS to VIIRS transition.*

c) Over ocean, median and 84th percentile values are also very similar between products. However, at 95th and 98th percentile events than VIIRS DB retrieves lower values-especially in the central Atlantic, high mid latitudes, and the Arctic. *Thus, over ocean we expect a decrease in AOD in the transition from MODIS to VIIRS for high AOD events.*

d) When comparing the number of days where AOD > 0.8, the contrast between the land and ocean further highlights the differences in land and ocean retrievals. The number of days where AOD > 0.8 also reveals more of the effects of swath width, sun glint, and different dust models used for ocean algorithms. When comparing the amount of joint detected 95th percentile AOD events against the single algorithm detected events, VIIRS DB has the highest likelihood of identifying 95th percentile events over ocean, while MODIS MAIAC presented the highest likelihood over land.

e) The global analysis was further investigated by comparing products at individual points through linear regression for AOD values < 0.8 and the mean bias for AOD > 0.8. Most notable is a clear reduction is slope from MODIS versus VIIRS, offset by a positive intercept. The highest coefficient of determination ($r^2$) is seen over coastal waters where there are dark ocean boundary conditions and higher AOD relative to open ocean. For land, the best correlations are seen over low albedo vegetated lands in biomass burning regions and pollution dominated regions. Lowest correlations are seen in regions with little dynamic range, such as the tropical Pacific Ocean, Chile, central Asia, western United States, and Australia. The strongest biases are seen for low AOD correlated with the issue of lower boundary conditions and at the high latitudes.

Based on the findings above, a series of more in-depth regional analysis was performed aimed to dissect the product differences in retrievals and sampling. This is done through comparisons to AERONET sensors, time series, probability distributions, and case studies. Regions investigated included a host of biomass burning, arid, polluted, and mixed environments.

a) The biomass burning dominated regimes were separated into Boreal Asia, Boreal Canada, Central Africa, and South America. All three satellite products are in relatively good agreement with one another and AERONET. These regions show that time series regional averaging can provide a good estimate of severe aerosol events especially where there is a good distribution of AERONET sites, such as in South America, whereas Boreal Asia and central Africa are underrepresented. Nevertheless, there is evidence of issues associated with sampling, scattering angle, the fine-coarse mode partitions, and under sampling of severe events. Algorithms intercompare best for South America and Central Africa, although MAIAC does exhibit a noticeable low bias. Boreal smoke is more problematic, with reversals in bias between algorithms between Asian and North American Boreal smoke plumes.



b) In the dust dominated Saharan region, MODIS products are lower than VIIRS due to changes in assumptions in dust optical properties. Higher values can also be associated with algorithm improvements in cloud and dust discrimination. There are few AERONET sites available for evaluation, however, and correlations are in determinant.

c) The last aerosol regions studied includes mixed pollution and dust regions within Asia. These regions include southwest Asia, Southeast Asia, south Asia, and eastern Asia. The correlations between products are strongest over land for dark vegetated surfaces and biomass burning in northern Southeast Asia and Eastern Asia. Like over the Sahara, lower correlations are exhibited in bright deserts. Given the mixed aerosol sources in these regions, there are often difficulties in the fine and coarse mode partition as well as the land/ocean boundaries. Sampling bias seems to occur based on the sensor locations.

This studies evaluation results show that even after 20 years of experience with dark target types of algorithms, correlated divergence between products is still problematic for higher fidelity applications and notably here, higher optical depths when multiple scattering aggravates errors in assumed aerosol optical properties. This will no doubt require adjustments in 2023 with the shift from the EOS to the JPSS constellations. It can be argued that the spatially correlated biases observed between products here is a natural result of the underdetermined observations that single view and non-polarization passive remote sensing provide for aerosol characterization. Nevertheless, until some agreed upon baseline is made in the community, scientific results on climate change, inverse modeling of sources, and aerosol impacts will continue to have regional biases. Next generation polarimeters are expected to provide additional information that is hoped to resolve regional biases that are observed in MODIS and VIIRS. Nevertheless, they too will require studies such as conduced here that in turn will require multi-year datasets for evaluation and algorithm integration.

**6 Code availability**

Publicly available software was used to produce the results in this paper. The L3 software used in the analysis is available in GitHub and is in the process of being open sourced.

**7 Data availability**

All datasets are publicly available through the NASA data centers or individual NASA products.

**8 Author contributions**

AG, JSR, and REH participated in the preparation of this manuscript and performed data analysis. NCH, RCL, JZ, and TFE provided expert advice and feedback on data products. PV is the author of the gridding software used in this study.

**9 Competing interests**



Some authors are members of the editorial board of journal Atmospheric Measurement Techniques. The peer-review
820    process was guided by an independent editor, and the authors have also no other competing interests to declare.

## 10 Acknowledgements

This project was funded by the Office of Naval Research Code 322. It is recognized that the development of an
algorithm, requires a team effort for development and verification across many organizations. We gratefully
825    acknowledge the University of Wisconsin SIPS and Goddard Space Flight Center LAADS for the processing and
archiving of VIIRS and MODIS data, used in this study. We also wish to thank the members of the federated Aerosol
Robotic Network (AERONET) program for their steadfast contributions to satellite product verification. We
acknowledge the use of imagery from the NASA Worldview application (https://worldview.earthdata.nasa.gov/), part
of the NASA Earth Observing System Data and Information System (EOSDIS).

830

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
