# Peer review of "Assessment of Severe Aerosol Events from NASA MODIS and VIIRS Aerosol Products for Data Assimilation and Climate Continuity"

_Atmospheric Measurement Techniques, 2022_

## Referee Comment (RC1)

**Review of** amt-2022-290, "Assessment of Severe Aerosol Events from NASA MODIS and VIIRS Aerosol Products for Data Assimilation and Climate Continuity", by Amanda Gumber, Jeffery S. Reid, Robert E. Holz, Thomas F. Eck, N. Christina Hsu, Robert C. Levy, Jianglong Zhang, and Paolo Veglio

The paper provides an assessment of two MODIS AOD datasets (the combined Dark Target / Deep Blue and MAIAC) in comparison to the operational VIIRS dataset (and AERONET ground-based sun photometer measurements) with a focus on high AOD events.

The paper is of high relevance as it provides significant and detailed insights into the performance and deviations of MODIS vs. VIIRS which prepare the ground for the transition between those instruments in late 2023. Given the importance of MODIS for operational modeling systems through data assimilation, the assessment of the consistency of continuing observations (e.g. for air quality) and of long-term Climate Data records after this transition is of high importance.

I found one type of maps presented in a confusing way / not completely described (mean bias – absolute or relative) – this should be harmonized and its descriptions extended for better clarity.

I have a few minor suggestions on wording and graphics, symbols where I struggled to grasp the key messages (see detailed comments at the end).

I therefore recommend publication with minor revisions.

Response to review questions

1.  Does the paper address relevant scientific questions within the scope of AMT?

    Yes, a specific assessment of high AOD cases and of the consistency od datasets related to a sensor transition are of great value and importance.

2.  Does the paper present novel concepts, ideas, tools, or data?

    Yes, the paper is innovative with its specific focus on a comprehensive assessment of high AOD cases for different regions stratified by their different dominant aerosol types or mixtures of them.

3.  Are substantial conclusions reached?

    Yes, the paper does go into detailed regional / aerosol-type stratified analysis, and in its conclusions extracts major overall findings relevant for data assimilation and direct use of VIIRS AOD in sequence to MODIS AOD.

4.  Are the scientific methods and assumptions valid and clearly outlined?

    Yes, the paper applies an appropriate combination of methods to assess / compare the long tails of probability distributions with their specific complexity of low

numbers. Applying different methodology for the bulk of the AOD range (< 0.8) and for the rare high AOD cases is fully appropriate. Furthermore, the limitations entailed in the methodology used as well as due to the Aeronet reference data limited coverage are clearly described to put this part of the analysis into the right context.

5.  Are the results sufficient to support the interpretations and conclusions?

    Yes, the paper does provide a wealth of plots which justify in detail the conclusions drawn; in particular in the analysis of specific regions in section 4. The discussions do include some qualitative aspects of possible reasons for the differences (as proving each hypothesis would go far beyond the paper's scope). However, those discussions are all well underpinned by plausible arguments based on the observations in the various plots.

    I found one type of maps presented in a confusing way / not completely described (mean bias – absolute or relative) – this should be harmonized and its descriptions extended for better clarity.

6.  Is the description of experiments and calculations sufficiently complete and precise to allow their reproduction by fellow scientists (traceability of results)?

    Yes, the methodology is either described or referenced in their underlying publications.

7.  Do the authors give proper credit to related work and clearly indicate their own new/original contribution?

    Yes, I could not identify any major gap in the references quoted and there are clear (and valid) statements highlighting the unique elements of the paper).

8.  Does the title clearly reflect the contents of the paper?

    Yes, the title highlights the focus on high AOD cases as well as the intended applications of data assimilation and climate data record consistency.

9.  Does the abstract provide a concise and complete summary?

    Yes, the abstract summarizes the goal, methodology and overall conclusions.

10. Is the overall presentation well structured and clear?

    Yes, the presentation starts with a brief introduction of the datasets / their underlying algorithms, and then conducts the analysis first globally and then regionally, which provides a rich set of analysis details.

11. Is the language fluent and precise?

    Yes, overall, the paper is well written. I have a few suggestions for small improvements.

12. Are mathematical formulae, symbols, abbreviations, and units correctly defined and used?

Yes, overall, I see clear definitions and consistent usage. I have a few minor suggestions for improvements (see below).

13. Should any parts of the paper (text, formulae, figures, tables) be clarified, reduced, combined, or eliminated?

No, as the authors have already taken care to split some further analysis into a supplement, so that the overall flow of the argumentation can be better followed.

14. Are the number and quality of references appropriate?

Yes, I see all relevant work cited.

15. Is the amount and quality of supplementary material appropriate?

I appreciate the provision of further interesting detail in the supplement.

My few small concerns

1) Maps of mean bias are presented in Fig. 5 (right column) and Fig. 6 (all) and only the caption of Fig. 5 calls it "mean relative bias". As there are no negative values and in Fig. 6 over ocean for the lowest AOD range the value 1.0 dominates, it cannot be absolute biases, but the captions / text on a fast reading made me expect absolute bias. Can you please add one or two sentences clearly defining what is shown in the maps and use consistent terminology to ease a reader's understanding?
2) In the abstract and the methodology you call your common consistent product used for the analysis "a Level3 product". I understand that you want to stress the consistent aggregation (which is important) and the fact, that you work with a gridded dataset. As there exist the operational Level3 products, can you please introduce a naming which makes it clear that you are using a specific gridded product which also included AERONET data and all satellite datasets under investigation which differs from (some of) the operational Level3 products?

Minor / detailed comments which could help optimize the paper for reading

- In a few places I got lost whether analysis over ocean is part of the paper or not
- You have many "-" between parts of sentences without empty spaces, which confuse reading – please change them all to " – "
- use "AERDB" instead of "DB" for VIIRS in all places to distinguish from the MODIS DB product
- There are a few cases, when your sentences are missing a verb (e. g. lines 35, 123, 575)
- Can you be consistent in "84th percentile …" instead of sometimes using "84th% …"?

- Please use "1° x 1°" consistently throughout the paper instead of "1 degree x 1 degree"
- Please use "biomass burning" consistently instead of sometimes only using "buring"
- Line 60 "conditions" should be moved before the bracket (then you keep together "lower boundary conditions")
- Lines 93-96: split into two sentences
- Line 107: AERONET is not a sensor (CIMEL is)
- Line 115: I would add "comparison of the" before "product performance"
- Line 128: add "to" before "provide"
- Line 129: can you find a more appropriate word to replace "finalizing"
- Line 132: I do not understand what you want to say with "use the benchmark AOD values" / line 576
- Line 133: can you explain the principle behind "designed after commonly applied DA products"
- Line 135: can you please say, which fraction of pixels is typically kept for each of the three dataset by applying the QA flags?
- Line 143: please add "as level2" after "criterion"
- Line 145: why do you use Aeronet lv1.5?
- Line 148: replace "isolate" by "separate"
- Line 163: which version of MYD04 do you analyse (collection 6.1 I guess) – please add.
- Line 163: what is the impact of comparing MCD19 which is a combined TERRA / AQUA product to the other afternoon only products?
- Line 167: it would be very interesting to include VIIRS /DT, but I accept it was not yet available when the study was made – consider to delete the sentences on the new VIIRS version (here and later in the paper)
- Line 170: this sentence says that there is no fine mode AOD over land for any of the products – please reword
- Sections 2.2.1 – 2.2.3: the MDOIS part is much longer than the other two – maybe consider to harmonize
- Line 179:  add "pre-defined" before "fine and coarse"
- Line 190: add "in visible bands" after "signal"
- Line 225: please explain "M band channels"
- Line 228/229: so VIIRs AERDB has fine mode over bright land?
- Line 245: what do you mean by "aggregated to longer time domains"
- Line 264: replace "this subsection" by "section 3"
- Caption to fig. 2: filtering of ratios for $84^{th}$ percentile  of AOD<0.1 (for one of the datasets or for all?
- Fig. 2: To simplify reading, I suggest a shorter legend title for the top right maps ($\sigma_g$) – better to define "$\sigma_g$ = AOD $84^{th}$ percentile / median AOD" in the text as equation
- Fig. 3: is the same filtering applied for rations as in fig. 2?
- Table 1 / caption: I would add "AOD550: " at the caption start
- Line 316: what does "good" mean here?

- Lines 365ff: I suggest to avoid the word "capture" as it indicates that an algorithm detects something as compared to a truth, which you explicitly say cannot be assessed. Better use "observe" or "detect" or similar.
- Lines 370 – 372: I find this a very complicated sentence to understand.
- Fig. 4 / caption: duplication "detection" – "detected"
- Line 406: better say "pre-defined" instead of "programmed
- Line 437: I would replace "excellent" by "outstanding"
- Line 437: what do you mean by "model comparisons"?
- Line 443: add "truly" before "different"
- Figure 7: the lines are for which dataset? Aeronet, I guess – please add
- Line 487: add "which" before "have"
- Line 514: replace "less different" by "smaller"
- Line 338: add "showed that they" before "were"
- Line 545: what do you mean by "observed integer factors"?
- Line 560: "constancy ???
- Line 562: replace "concerning" by "of concern"
- Line 573f: "Notable are differences include" is not a proper sentence.
- Line 629: replace "lower still" by "even lower"
- Line 725: replace "particulate" by "particular"
- Line 746: delete "when"
- Line 771: sometimes you use past tense, mostly you use present tense.
- Line 780: "under-sampling"
- Line 788: delete "The last aerosol regions studied" by combining the first two sentences "Regions with mixed pollution and dust within Asia include southwest Asia …"
- Line 794: is "dark target types of algorithms" the correct summarizing of all algorithms here?

---

## Author Comment (AC1)

**Response to Review #1**

Review of amt-2022-290, "Assessment of Severe Aerosol Events from NASA MODIS and VIIRS Aerosol Products for Data Assimilation and Climate Continuity", by Amanda Gumber, Jeffery S. Reid, Robert E. Holz, Thomas F. Eck, N. Christina Hsu, Robert C. Levy, Jianglong Zhang, and Paolo Veglio

The paper provides an assessment of two MODIS AOD datasets (the combined Dark Target / Deep Blue and MAIAC) in comparison to the operational VIIRS dataset (and AERONET ground-based sun photometer measurements) with a focus on high AOD events.

The paper is of high relevance as it provides significant and detailed insights into the performance and deviations of MODIS vs. VIIRS which prepare the ground for the transition between those instruments in late 2023. Given the importance of MODIS for operational modeling systems through data assimilation, the assessment of the consistency of continuing observations (e.g. for air quality) and of long-term Climate Data records after this transition is of high importance.

I found one type of maps presented in a confusing way / not completely described (mean bias – absolute or relative) – this should be harmonized and its descriptions extended for better clarity.

I have a few minor suggestions on wording and graphics, symbols where I struggled to grasp the key messages (see detailed comments at the end).

I therefore recommend publication with minor revisions.

Response to review questions
1. Does the paper address relevant scientific questions within the scope of AMT?
Yes, a specific assessment of high AOD cases and of the consistency od datasets related to a sensor transition are of great value and importance.

2. Does the paper present novel concepts, ideas, tools, or data?
Yes, the paper is innovative with its specific focus on a comprehensive assessment of high AOD cases for different regions stratified by their different dominant aerosol types or mixtures of them.

3. Are substantial conclusions reached?
Yes, the paper does go into detailed regional / aerosol-type stratified analysis, and in its conclusions extracts major overall findings relevant for data assimilation and direct use of VIIRS AOD in sequence to MODIS AOD.

4. Are the scientific methods and assumptions valid and clearly outlined?
Yes, the paper applies an appropriate combination of methods to assess / compare the long tails of probability distributions with their specific complexity of low numbers. Applying different methodology for the bulk of the AOD range ($< 0.8$) and for the rare high AOD cases is fully appropriate. Furthermore, the limitations entailed in the methodology used as well as due to the

Aeronet reference data limited coverage are clearly described to put this part of the analysis into the right context.

5. Are the results sufficient to support the interpretations and conclusions?
Yes, the paper does provide a wealth of plots which justify in detail the conclusions drawn; in particular in the analysis of specific regions in section 4. The discussions do include some qualitative aspects of possible reasons for the differences (as proving each hypothesis would go far beyond the paper's scope). However, those discussionsare all well underpinned by plausible arguments based on the observations in the various plots.
I found one type of maps presented in a confusing way / not completely described (mean bias – absolute or relative) – this should be harmonized and its descriptions extended for better clarity.

6. Is the description of experiments and calculations sufficiently complete and precise to allow their reproduction by fellow scientists (traceability of results)?
Yes, the methodology is either described or referenced in their underlying publications.

7. Do the authors give proper credit to related work and clearly indicate their own new/original contribution?
Yes, I could not identify any major gap in the references quoted and there are clear (and valid) statements highlighting the unique elements of the paper).

8. Does the title clearly reflect the contents of the paper?
Yes, the title highlights the focus on high AOD cases as well as the intended applications of data assimilation and climate data record consistency.

9. Does the abstract provide a concise and complete summary?
Yes, the abstract summarizes the goal, methodology and overall conclusions.

10. Is the overall presentation well-structured and clear?
Yes, the presentation starts with a brief introduction of the datasets / their underlying algorithms, and then conducts the analysis first globally and then regionally, which provides a rich set of analysis details.

11. Is the language fluent and precise?
Yes, overall, the paper is well written. I have a few suggestions for small improvements.

12. Are mathematical formulae, symbols, abbreviations, and units correctly defined and used?
Yes, overall, I see clear definitions and consistent usage. I have a few minor suggestions for improvements (see below).

13. Should any parts of the paper (text, formulae, figures, tables) be clarified, reduced, combined, or eliminated?
No, as the authors have already taken care to split some further analysis into a supplement, so that the overall flow of the argumentation can be better followed.

14. Are the number and quality of references appropriate?

Yes, I see all relevant work cited.

15. Is the amount and quality of supplementary material appropriate?
I appreciate the provision of further interesting detail in the supplement.

My few small concerns
**1) Maps of mean bias are presented in Fig. 5 (right column) and Fig. 6 (all) and only the caption of Fig. 5 calls it "mean relative bias". As there are no negative values and in Fig. 6 over ocean for the lowest AOD range the value 1.0 dominates, it cannot be absolute biases, but the captions / text on a fast reading made me expect absolute bias. Can you please add one or two sentences clearly defining what is shown in the maps and use consistent terminology to ease a reader's understanding?**

**Response:** *Thank you for the suggestion. You are correct that it should be better defined. Rather than mean bias, it is the average of the ratios of the pairwise satellite products. In Fig. 6 over ocean where the value 1.0 dominates indicates that the products show small differences and have a relatively low bias. Both Figure 5 & 6 have been updated to read, "Ratio of AOD > 0.8" and "Pairwise Mean Ratio." We added a sentence to clarify this at line 492 in the revised manuscript: "The mean ratios calculated in Fig. 5 and Fig. 6 are defined as the averages of the ratios of the pairwise satellite products."*

**2) In the abstract and the methodology you call your common consistent product used for the analysis "a Level3 product". I understand that you want to stress the consistent aggregation (which is important) and the fact, that you work with a gridded dataset. As there exist the operational Level3 products, can you please introduce a naming which makes it clear that you are using a specific gridded product which also included AERONET data and all satellite datasets under investigation which differs from (some of) the operational Level3 products?**

**Response***: Thanks for the recommendation. We have given the product a name, the SSEC/NRL L3 product and have added that product name throughout the revised manuscript. We have also added the sentence at line 255: "The creation of this SSEC/NRL L3 product differs from other operational L3 products because of the use of a consistent aggregation method and the options of filtering and masking which Yori provides."*

**Minor / detailed comments which could help optimize the paper for reading**
**- In a few places I got lost whether analysis over ocean is part of the paper or not**

**Response:** *For the global analysis (Section 4), we do include both land and ocean. For the regional analysis (Section 4), we chose to focus on land only for comparison with AERONET. We added a reference to this in the revised manuscript at line 507: "For better comparison to AERONET, this regional analysis will focus only on land data."*

**- You have many "-" between parts of sentences without empty spaces, which confuse reading – please change them all to " – "**
**Response:** *Noted and updated.*

**- use "AERDB" instead of "DB" for VIIRS in all places to distinguish from the MODIS DB Product**
**Response:** *Noted and updated.*

**- There are a few cases, when your sentences are missing a verb (e. g. lines 35, 123, 575)**
**Response:** *Noted and updated*

**- Can you be consistent in "84th percentile ..." instead of sometimes using "84th% ..."?**
**Response:** *Noted and updated*

**- Please use "1° x 1°" consistently throughout the paper instead of "1 degree x 1 degree"**
**Response:** *Noted and updated*

**- Please use "biomass burning" consistently instead of sometimes only using "burning"**
**Response:** *Noted and updated*

**- Line 60 "conditions" should be moved before the bracket (then you keep together "lower boundary conditions")**
**Response:** *Noted and updated*

**- Lines 93-96: split into two sentences**
**Response:** *Noted and updated*

**- Line 107: AERONET is not a sensor (CIMEL is)**
**Response:** *Changed sensor to instrument*

**- Line 115: I would add "comparison of the" before "product performance"**
**Response:** *Noted and updated*

**- Line 128: add "to" before "provide"**
**Response:** *Noted and updated*

**- Line 129: can you find a more appropriate word to replace "finalizing"**
**Response:** *Changed finalized to resolved*

**- Line 132: I do not understand what you want to say with "use the benchmark AOD values" / line 576**
**Response:** We *deleted the word "benchmark," it was intended to describe the most studied wavelength of AOD at 550nm.*

**- Line 133: can you explain the principle behind "designed after commonly applied DA products"**

**Response:** *It was intended to describe how many data assimilation products are set to a 1x1 degree grid. We changed the phase to "designed after commonly applied DA products" to "similar to commonly applied DA products"*

- Line 135: can you please say, which fraction of pixels is typically kept for each of the three dataset by applying the QA flags?
**Response:** *A sentence at line 138 has been added to address the fraction of pixels kept using the QA flags: "By using the highest quality retrievals, the amount of AOD pixels is reduced by approximately 55%, 52%, and 51% for VIIRS AERDB, MODIS DT/DB and MODIS MAIAC."*

**- Line 143: please add "as level2" after "criterion"**
**Response:** *Noted and updated*

**- Line 145: why do you use Aeronet lv1.5?**
**Response:** *When the dataset for this study was produced, some sites were still only AERONET Level 1.5 at the tail end of the study period. When a site elevates officially from 1.5 to 2, the 1.5 data is replaced by level 2 within the file.*

**- Line 148: replace "isolate" by "separate"**
**Response:** *Noted and updated*

**- Line 163: which version of MYD04 do you analyse (collection 6.1 I guess) – please add.**
**Response:** *Noted and updated*

**- Line 163: what is the impact of comparing MCD19 which is a combined TERRA / AQUA product to the other afternoon only products?**
**Response:** *There is a flag in the product to separate Aqua and Terra, so only Aqua is used in the study. For clarity, we added the sentence, "The product can be separated by satellite, so only Aqua is used in this study." at line 176.*

**- Line 167: it would be very interesting to include VIIRS /DT, but I accept it was not yet available when the study was made – consider to delete the sentences on the new VIIRS version (here and later in the paper)**
**Response:** *We think it is important to acknowledge why VIIRS DT is not in this study especially since it is a currently available dataset now.*

**- Line 170: this sentence says that there is no fine mode AOD over land for any of the products – please reword**
**Response:** *In revising the manuscript, this sentence has been removed to shorten the section and because fine mode fraction was not used in the results of this study.*

**- Sections 2.2.1 – 2.2.3: the MDOIS part is much longer than the other two – maybe consider to harmonize**

**Response***: Thanks for the recommendation, we did try to shorten the MODIS Combined DT/DB section in the revised manuscript. We think it does end up needing to be a little longer than the others given that it is a combination of multiple algorithms.*

**- Line 179: add "pre-defined" before "fine and coarse"**
**Response:** *Noted and updated*

**- Line 190: add "in visible bands" after "signal"**
**Response**: *Noted and updated*

**- Line 225: please explain "M band channels"**
**Response***:The M stands for moderate resolution. We added in "moderate resolution" for in front of M band in line 234*

**- Line 228/229: so VIIRs AERDB has fine mode over bright land?**
**Response:** *Thank you for pointing this out. It does not have fine mode over land, and we altered sentence to say, "their properties" instead of "fine mode fraction" in the revised manuscript.*

**- Line 245: what do you mean by "aggregated to longer time domains"**
**Response***: The phrase "aggregated to longer time domains" was supposed to mean they were created into daily files. This sentence has since been removed from the revised manuscript.*

**- Line 264: replace "this subsection" by "section 3"**
**Response**: *Noted and updated*

**- Caption to fig. 2: filtering of ratios for 84th percentile of AOD<0.1 (for one of the datasets or for all?**
**Response***: Yes, the ratios are filtered for the 84$^{th}$ percentile of AOD<0.1 for all datasets. We chose to do this filtering to enhance regions of interest for severe aerosol events.*

**- Fig. 2: To simplify reading, I suggest a shorter legend title for the top right maps (σg) – better to define "σg = AOD 84th percentile / median AOD" in the text as equation**
**Response**: *Noted and updated*

**- Fig. 3: is the same filtering applied for rations as in fig. 2?**
**Response**: *No, the same filtering was not applied to the ratios in fig 3. The filter was used in Fig 2. to remove areas that don't experience high aerosol loading. In fig. 3, the filter was not used to look at 95$^{th}$ and 98$^{th}$ percentile events.*

- Table 1 / caption: I would add "AOD550: " at the caption start
**Response**: *Noted and updated*

**- Line 316: what does "good" mean here?**
**Response**: *"AOD 550 signal is good" refer to the signal being good at revealing areas of seasonal aerosol loading. Added the text, "in revealing areas of seasonal aerosol loading" after the word good.*

**- Lines 365ff: I suggest to avoid the word "capture" as it indicates that an algorithm detects something as compared to a truth, which you explicitly say cannot be assessed. Better use "observe" or "detect" or similar.**
**Response**: *Changed "capture" to "observes"*

**- Lines 370 – 372: I find this a very complicated sentence to understand.**
**Response**: *Sentence has been re-worded to be hopefully more understandable at line 405*

**- Fig. 4 / caption: duplication "detection" – "detected"**
**Response**: *Noted and updated*

**- Line 406: better say "pre-defined" instead of "programmed**
**Response**: *Noted and updated*

**- Line 437: I would replace "excellent" by "outstanding"**
**Response**: *Noted and updated*

**- Line 437: what do you mean by "model comparisons"?**
**Response:** *Thank you for pointing that out. By model comparisons, we mean to reference the article Reid et al. (2022) where the ICAP consensus models are compared to MODIS. These models do not have the sharp boundaries for significant land plume ejections like the MODIS product does. This provides evidence that there are differences between the land and ocean retrievals resulting in coastal changes in AOD even when compared pairwise. We have added this description to line 474.*

**- Line 443: add "truly" before "different"**
**Response**: *Noted and updated*

**- Figure 7: the lines are for which dataset? Aeronet, I guess – please add**
**Response***: The lines are a combination of all the datasets defined in the figure caption.*

**- Line 487: add "which" before "have"**
**Response**: *Noted and updated*

**- Line 514: replace "less different" by "smaller"**
**Response**: *Noted and updated*

**- Line 538: add "showed that they" before "were"**
**Response:** *Noted and updated*

**- Line 545: what do you mean by "observed integer factors"?**
**Response:** *"Observed integer factors" is intended to describe the large amount of bias between products.*

**- Line 560: "constancy ???**

**Response:** *Constancy changed to consistency*

**- Line 562: replace "concerning" by "of concern"**
**Response:** *Noted and updated*

**- Line 573f: "Notable are differences include" is not a proper sentence.**
**Response***: We removed the word "are" to make it a proper sentence. Thank you for catching that error.*

**- Line 629: replace "lower still" by "even lower"**
**Response:** *Noted and updated*

**- Line 725: replace "particulate" by "particular"**
**Response:** *Noted and updated*

**- Line 746: delete "when"**
**Response***: Noted and updated*

**- Line 771: sometimes you use past tense, mostly you use present tense.**
**Response:** *Noted and updated*

**- Line 780: "under-sampling"**
**Response:** *Noted and updated*

**- Line 788: delete "The last aerosol regions studied" by combining the first two sentences "Regions with mixed pollution and dust within Asia include southwest Asia ..."**
**Response:** *Noted and updated*

**- Line 794: is "dark target types of algorithms" the correct summarizing of all algorithms here?**
**Response:** *Yes, we would say it is correct. The Dark Target algorithm is one of the longer studied algorithms and has been expanded upon and incorporated into both MODIS and VIIRS products.*

---

## Author Comment (AC2)

**Response to Review #2**

Review of "Assessment of Severe Aerosol Events from NASA MODIS and VIIRS Aerosol Products for Data Assimilation and Climate Continuity"

General comments

The objective of this paper is to assess the differences between MODIS-based and VIIRS AOD products. In the context of a likely disruption of the MODIS product, assessing and documenting the consistencies and inconsistencies between MODIS and VIIRS data sets is essential for AOD data assimilation and to ensure the continuity of essential climate variable production. This paper provides a unique and meaningful documentation of the differences between products as well as their performances evaluated against AERONET at both global and regional scales. The analysis of the probability distribution function of each dataset provides a relevant statistical characterization on how the data sets compare each other in terms of capturing major aerosol events. The pairwise comparison informs on product differences for different AOD regimes that can be related to differences in each retrieval algorithm. Finally, the regional analysis allows to identify the product strengths and deficiencies for different regions. While the scientific contribution of this paper is strong and very relevant for its publication in AMT, several aspects of the paper, which are underlined below, should be improved prior to publication.

My main concern is about the description of methodology. There is no dedicated methodological section. The authors have chosen to separate the paper into the 3 types of analysis, namely: probability distribution function, regression and regional analysis, which include a brief method description along with the results and their interpretation. The author can keep that approach but should include a subsection dedicated to methodology in each analysis section or should consider having a separate section on method (similarly to the data one). Several aspects of the methodology should be better presented: what is the role of AERONET ? as far as I read it is involved in the regional analysis and not the global comparison ? Is it used as a reference data set in term of accuracy ? For the pairwise analysis, the method should be clearly explained, the reference to the past paper is not enough.

**Response**: *Thank you for your comments and review. We have added a methodology section (Section 3) to clarify what we are doing and how we use the data.*

Specific comments

- **The title is too long**-

**Response:** *Yes, we agree that it is a long title, but we do think it is to the point given the length of what we cover in the manuscript. We have not been able to come up with an alternate but are open to suggestions.*

- **Abstract: the role of AERONET is not clear, it is presented at same level as the satellite dataset but should be considered as a reference data set because of higher accuracy:**

**Response***: Thank you for pointing this out. The role of AERONET is considered the closest form of validation to the satellite datasets. This is now stated in Line 30 of the abstract. This said, as is noted in the manuscript AERONET data as point data is aliased. Therefore, we examine the three satellite observations all together.*

- **Introduction**
  - o For the sources of AOD retrieval uncertainty:
    - ▪ The measurement information content is a major source of uncertainties and it depends on geometry and the range of scattering angle which is sampled by the instrument
    - ▪ Cloud screening is a major source of uncertainty in aerosol retrieval and a large source of departures between products. This interacts in a complex manner with the differences in spatial resolution between products.

**Response**: *Yes, we agree that both of those are sources of AOD retrieval uncertainty. Section 5 addresses both of these uncertainties throughout the case studies in the regional analysis.*

- • Regarding the definitions given for data assimilation, observation error and bias: I found it confusing. DA aims at correcting only small amount of random error that can be quantified by the SD of the differences between the observation and its model-simulated equivalent. Observation and model first guess should be unbiased in theory. Bias correction scheme aims at removing any systematic differences between the observation and the model.

**Response:** *Regarding observation error and bias for data assimilation, we are referring to using only the highest quality of data as input to a data assimilation model. Zhang and Reid (2006) discuss the importance of this and how using biased retrievals in data assimilation affects the accuracy of both local and regional analysis. As data sources transition from MODIS and VIIRS products, it is important to identify the differences between products based on instrument and retrieval algorithm differences.*

  - o VIIRS AOD: It should be clearly acknowledged that there are two distinct datasets for VIIRS: one produced by NASA and one produced by NOAA

    **R***esponse: Thank you for bringing this point up. It is important to acknowledge that there are multiple products available. We included NASA in the title and added a reference to NASA in the section title, "2.2.3 VIIRS NASA Deep Blue." We also added the sentence, "There are two primary aerosol products produced for VIIRS by both NASA and NOAA, but this study only uses the NASA product." at line 260 in the revised manuscript.*

  - o The objective of the paper should be better explained.

    **Response:** *We think that the objective of the paper is best described in lines 110-132. With MODIS nearing retirement, there are going to be large changes required for those who use this data in the data assimilation community. Harmonization is going to be*

*required as data sources transition from MODIS to VIIRS products. The objective of this manuscript is to serve as a starting point of how these datasets currently monitor severe aerosol events.*

- The paper does not show/discuss the differences between product from same instrument but from distinct satellites: TERRA vs AQUA; S-NPP vs NOAA20. This is quite important in particular in the context of data assimilation when one product from one satellite can be biased due to radiometric uncertainties, I would suggest to include some results, if possible apply the intercomparison metrics separately to instrument and platforms.

  **Response:** *Yes, this is a great point to bring up. The purpose of this study was to focus on products that are in the same orbit or in close observation time to one another. Since Terra and Aqua are in a different orbit, so you can't do 1:1 matching between the two. To compare S-NPP and NOAA 20, there is not enough overlap during their operational periods. Such issues are being considered in ongoing work, but it is beyond the scope of this paper which is already long by most standards. Here we are making note of the nature of key differences, and that they can be quite large. We are working on bias correction methodologies that can be applied later.*

- **Satellite AOD (section 2): The description of AOD product is too long. The statements on MODIS instrument characteristics are not essential, the readers can refer to dedicated papers. Any references or statement on the differences between the NASA and NOAA products would be helpful. As well, I suggest to include a Table which summarize the main characteristics of each product and that would help to identify their differences. The YORI method is too detailed, please provide the essential information**

  **Response:** *Thank you for the comments on this section. Portions of the MODIS products descriptions have been shortened. The YORI method description has been edited and moved into the methodology section. We think it is important to emphasize the use of YORI given another reviewer has noted that we must stress how this product is different than other L3 products. We have also added clarity that we are using the NASA VIIRS product vs the NOAA VIIRS product as mentioned in the previous comment.*

- **Section 3.2 line 395-397: it is not clear what do the author mean with nonlineraties in AOD**

  **Response:** *It is meant to describe that at high AOD there is not often a direct relationship between the different datasets which makes using a linear regression less representative of the data. As AOD increases, it approaches optical depth semi-infinite and the ratio of radiance to AOD diminishes with increasing AOD. At that point intensive parameters like single scattering albedo become more important that AOD itself.*

- **Line 400-403: the sentence about dynamic range is not clear**

  **Response***: The sentence is referring to the resulting sampling bias of using correlation coefficients for regions where there is a large range of AOD vs low a low range of AOD environments. That is, for a simple error model (e.g., a+B*AOD), simply by having a larger dynamic range (a wider span of relative lower to higher values) the r2 value will improve. Thus, r2, as is commonly used as a benchmark, inherently penalizes regions of low AOD.*

- **Use mean deviation (MD) instead of bias for product comparison, bias is generally used with respect to reference measurements (such as AERONET)**

  **Response:** *Thank you for bringing up this point. It does make sense to use the terminology of mean deviation vs mean bias since we are comparing two unverified products. There have been changes made throughout the revised manuscript to reflect this comment*

- **Not enough analysis with respect to the impact of differences in cloud screening between products**

  **Response:** *Cloud screening is a large part of any aerosol algorithm so in studying the differences between satellite products we can see the impacts of cloud contamination in the L3 product we've created. An improved cloud bias analysis is underway and will appear in a separate paper. It is also important to note that using the highest quality flag for each of these does filter out a majority of the detected cloud fraction and these algorithms tend to lean clear sky conservative.*

- **Line 584-587: this statement is not specific to this paragraph**

  **Response:** *We respectively disagree and think that this statement is relevant to describing the list of differences described in the paragraph above.*